# Mitigating Spurious Features in Contrastive Learning with Spectral Regularization

**Naghmeh Ghanooni**
Department of Computer Science, RPTU
Kaiserslautern, Germany
`ghanooni@rptu.de`

**Waleed Mustafa***
Department of Computer Science, RPTU
Kaiserslautern, Germany
`mustafa@rptu.de`

**Dennis Wagner***
Department of Computer Science, RPTU
Kaiserslautern, Germany
`wagner@rptu.de`

**Anthony Widjaja Lin**
Max-Planck Institute for Software Systems
Kaiserslautern, Germany
`lin@cs.uni-kl.de`

**Sophie Fellenz**
Department of Computer Science, RPTU
Kaiserslautern, Germany
`fellenz@rptu.de`

**Marius Kloft**
Department of Computer Science, RPTU
Kaiserslautern, Germany
`kloft@rptu.de`

## Abstract

Neural networks generally prefer simple and easy-to-learn features. When these features are spuriously correlated with the labels, the network's performance can suffer, particularly for underrepresented classes or concepts. Self-supervised representation learning methods, such as contrastive learning, are especially prone to this issue, often resulting in worse performance on downstream tasks. We identify a key spectral signature of this failure: early reliance on dominant singular modes of the learned feature matrix. To mitigate this, we propose a novel framework that promotes a uniform eigenspectrum of the feature covariance matrix, encouraging diverse and semantically rich representations. Our method operates in a fully self-supervised setting, without relying on ground-truth labels or any additional information. Empirical results on SimCLR and SimSiam demonstrate consistent gains in robustness and transfer performance, suggesting broad applicability across self-supervised learning paradigms. Code: GitHub repository.

## 1 Introduction

Neural networks tend to prioritize learning simple and easily detectable features in the early stages of training before capturing more complex patterns, revealing an inherent simplicity bias in neural network optimization [Geirhos et al., 2020, Shah et al., 2020, Rahaman et al., 2019, Kalimeris et al., 2019, Xue et al., 2023]. While this inductive bias can accelerate learning, it also makes models vulnerable to relying on features that are only superficially correlated with the labels due to dataset-specific artifacts.

These features, known as *spurious features*, can dominate model behavior, since they are easier to learn than the true task-relevant signals [Sagawa et al., 2019, Qiu et al., 2024, Sagawa et al., 2020, Kirichenko et al., 2022, Murali et al., 2023]. For instance, Zech et al. [2018] showed that a model

---

*Equal contribution; authors listed alphabetically.

39th Conference on Neural Information Processing Systems (NeurIPS 2025).

trained to detect pneumonia in chest X-rays relied on visual artifacts—such as hospital-specific metal tokens—rather than medical markers of the disease. Although these shortcuts may lead to strong performance on training and validation distribution, they often fail to generalize to rarer or more challenging inputs. *Thus, models that appear accurate may perform poorly across the full spectrum of real-world variation.*

Recent work shows that self-supervised representation learning (SSRL) is not immune to spurious correlations either, even though trained on unlabeled data [Hamidieh et al., 2024, Zhu et al., 2023]. In particular, contrastive learning (CL; a popular technique in SSRL) learns features—also referred to as representations—that are intended to be broadly useful across a wide range of downstream tasks, by encouraging similarity between augmented views of the same instance [Chen et al., 2020, He et al., 2020, Grill et al., 2020, Chen and He, 2021, Caron et al., 2020, Zbontar et al., 2021]. However, CL's objective of maximizing agreement between views can lead the model to rely on patterns that are predictive within the training distribution but unreliable in unfamiliar or varied conditions. In doing so, CL may overfit to spurious signals present in the training data, ultimately limiting the generalization of the learned representations to novel or diverse settings.

Whether a feature is considered spurious depends on the downstream task. A features that is irrelevant—or even misleading—for one task may be essential for another. This task-dependence highlights a key challenge in self-supervised learning: *the absence of knowledge about the downstream task during pretraining*. Without this information, task-relevant features (often referred to as *core features*) and spurious features are indistinguishable from the perspective of the unlabeled training data. As a result, self-supervised methods can entangle both in their learned representations, limiting their robustness and transferability. Despite progress in SSLR, we lack a theoretical understanding and practical solution for how to mitigate the influence of spurious signals during self-supervised pretraining.

To evaluate the quality of the learned features, it is common practice to assess how well simple linear classifiers perform on downstream tasks. Which raises the question, whether we can train a single, general-purpose representation that enables a wide range of downstream tasks to be solved with nothing more than a shallow model? We explore this question by theoretically investigating why standard networks tend to prefer spurious features. We show that the generalization performance of a downstream task closely tied to the spectrum of the matrix $XX^\top$, where $X$ is the input feature matrix. Dimensions associated with larger eigenvalues are learned first. Since spurious features are typically simple and easy to learn, they dominate the top eigenspectrum, leading the model to focus on them and reinforce them disproportionately.

To counteract this tendency, we introduce a *rank-promoting regularizer* that flattens the spectrum of $XX^\top$. By lifting the smaller eigenvalues and reducing their disparity with the largest ones, the method encourages the model to allocate capacity more uniformly across all informative directions, promoting the learning of both core and spurious factors. Importantly, the regularizer is architecture-agnostic and can be seamlessly integrated into any SSRL pipeline without modification.

The main contributions of this paper are summarized as follows:

- We introduce a synthetic dataset that elucidates why neural networks tend to prioritize spurious features over core features, and demonstrate—via rank analysis—how such preferences reduce representational diversity during training.

- We provide a theoretical analysis showing that generalization in downstream tasks depends on the eigenvalue spectrum of $XX^\top$, with uniform spectra yielding optimal transferability.

- Based on this insight, we propose a simple, architecture-agnostic regularizer that flattens the spectrum of $XX^\top$, encouraging diverse representations. The regularizer integrates seamlessly into any SSLR with negligible overhead.

- Experiments across five challenging spurious-correlation benchmarks show that the method substantially improves performance on the most difficult portion of the dataset—worst-group accuracy— while also increasing average accuracy, and achieves new state-of-the-art results on multiple downstream tasks.

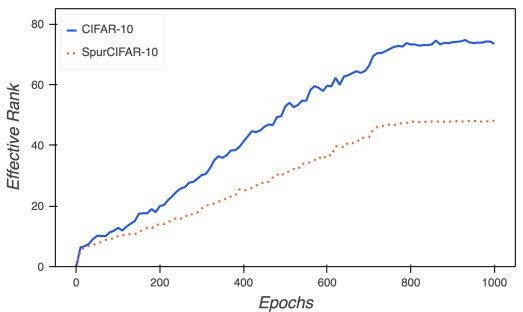 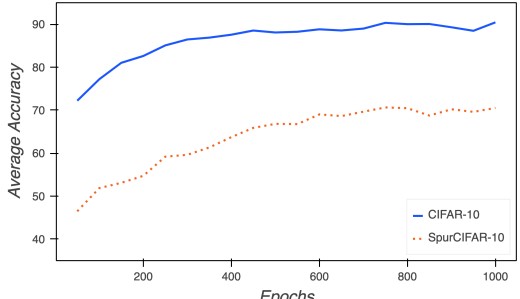

(a) The effective rank is lower when training on CIFAR-10 with artificial spurious features.

(b) Classification performance on CIFAR-10 is worse in the presence of artificial spurious features.

Figure 1: Training a classifier on CIFAR-10 with artificially added, strongly correlated spurious features results in noticeably worse performance and a lower effective rank of the learned representations, suggesting a potential connection between representation diversity and generalization.

## 2 Spectral Imbalance Limits Downstream Flexibility

Neural networks are known to prioritize simple, easily learnable features during training [Geirhos et al., 2020, Rahaman et al., 2019, Xue et al., 2023]. In self-supervised pretraining, this often leads to the dominance of spurious features—those that are easy to align between augmented views but are semantically irrelevant and fail to generalize beyond the training distribution. These dominant components shape the spectral structure of the learned representation space. In this section, we show how such spectral imbalance constrains downstream learning, and how a more uniform spectrum enables better task adaptation.

### 2.1 Gradient Flow and Spectral Bias

To understand how spectral structure biases downstream learning, we analyze the optimization dynamics of a linear predictor trained on frozen representations from a pretrained encoder. Let $f(\cdot)$ denote a feature representation. Given a training set $\{x_i\}_{i=1}^n$, define the corresponding feature matrix as $F = [f(x_1), \ldots, f(x_n)]^\top \in \mathbb{R}^{n \times d}$. We train a linear predictor $g_\mathbf{w}(f(\mathbf{x})) = \langle f(\mathbf{x}), \mathbf{w} \rangle$ using gradient flow on a loss $\Phi$:

$$\frac{d\mathbf{w}}{dt} = -\frac{d\Phi}{d\mathbf{w}}, \quad \text{so} \quad \frac{dg}{dt} = -FF^\top \frac{d\Phi}{dg} = -\sum_{i=1}^n \lambda_i v_i v_i^\top \frac{d\Phi}{dg},$$

where $FF^\top = \sum_{i=1}^n \lambda_i v_i v_i^\top$ is the eigendecomposition of the feature covariance matrix.

This shows that the evolution of predictions is dominated by top eigendirections of $FF^\top$, as those with larger eigenvalues accelerate learning. Consequently, downstream predictors are biased toward dominant spectral directions—regardless of whether they encode meaningful or spurious information. Since spurious features are often the easiest to learn during pretraining, they are likely to occupy top eigenspaces, constraining downstream learning to brittle, semantically weak directions.

To quantify spectral imbalance, we use the *effective rank*, a principled measure based on the entropy of the singular value spectrum:

**Definition 1** (Effective Rank [Roy and Vetterli, 2007]). *Let $A \in \mathbb{R}^{n \times d}$ have singular values $\sigma_1, \ldots, \sigma_r$, with $r = \min(n, d)$. Define:*

$$p_k = \frac{\sigma_k}{\sum_{j=1}^r \sigma_j}, \quad \text{rank}^{eff}(A) = \exp\left(-\sum_{k=1}^r p_k \log p_k\right).$$

A high effective rank indicates a flatter spectrum and greater diversity of informative directions. A low effective rank implies that variance is concentrated in a narrow subspace—often aligned with spurious features—limiting downstream flexibility.

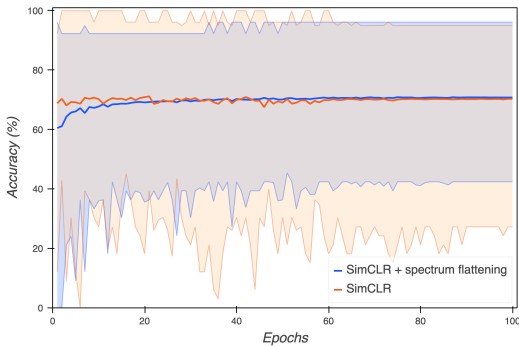 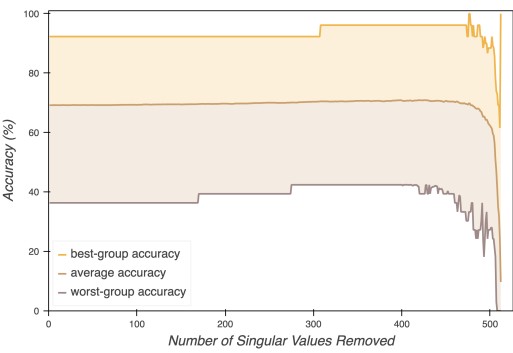

(a) Dropping the 400 smallest singular values and flattening doubles worst-group accuracy (20 → 40 %).

(b) Removed (0 to 512) with the remaining spectrum flattened. Progressive truncation improves robustness.

Figure 2: Impact of spectral interventions on SimCLR-trained features for SpurCIFAR-10. Removing low-energy directions and flattening the remaining spectrum can double worst-group accuracy without any architectural changes. Shaded bands around each curve represent worst- and best-group accuracy. Left: single 400-mode cut; right: incremental cuts.

## 2.2 Spurious Features Collapse the Spectrum

We illustrate the effect of spectral imbalance by comparing SimCLR-trained representations on CIFAR-10 and SpurCIFAR-10, the latter introducing a spurious correlation of 0.95 via class-dependent overlays. Using a ResNet-18 encoder (512 dimensions, 1000 epochs), we compute the effective rank of the training feature matrix and evaluate test accuracy on 5,000 held-out examples.

As shown in Figure 1, the effective rank of the representation matrix is significantly lower on SpurCIFAR-10, accompanied by reduced test accuracy. This confirms that spurious features lead to a collapsed spectrum, reducing representational diversity and flexibility.

These observations support our central hypothesis:

**Hypothesis 2** (Spectral Diversity and Learnability). *The number of significant singular values in the feature matrix governs the learnability of downstream tasks. Representations with higher effective rank are more likely to support robust generalization across diverse tasks.*

## 2.3 Spectral Interventions and Robustness

To test this hypothesis, we manipulate the spectrum of the feature matrix learned by SimCLR on SpurCIFAR-10. Specifically, we truncate the smallest 400 singular values and flatten the remaining spectrum. A linear classifier trained on this modified representation (Figure 2a) shows a substantial gain in worst-group accuracy—from approximately 20% to 40%—while maintaining or slightly improving average accuracy and the best-group accuracy. Figure 2b shows that progressively removing small singular values yields only modest, incremental gains in worst-group performance. Taken together, the findings indicate that low-rank directions offer only marginal benefit to generalization and can even slightly undermine robustness. To isolate the effect of flattening, we conduct an ablation study where only the smallest singular values are removed without flattening the remaining spectrum. As shown in Appendix A, Figure 3, this approach also yields significant improvement in worst-group accuracy, showing the importance of a balanced spectrum for robustness.

Our findings complement prior work on the role of rank in deep learning [Sainath et al., 2013, Feng et al., 2022, Andriushchenko et al., 2023], while highlighting a different trade-off: contrastive learning benefits not from aggressive compression but from maintaining a balanced spectrum. Even in a 512-dimensional space, the effective rank remains relatively low; increasing it improves robustness. Importantly, our goal is not to eliminate spurious features—some may benefit certain tasks—but to ensure that task-relevant signals are preserved. Flattening the spectrum promotes richer, more flexible representations that enable robust generalization—even in the presence of spurious correlations.

# 3 Related Work

**Spurious Correlations and Robustness**    Several studies have examined how neural networks rely on spurious features [Geirhos et al., 2020, Sagawa et al., 2020, Shah et al., 2020]. For instance, Zech et al. [2018] showed that models trained on chest X-rays often rely on hospital-specific artifacts rather than medical content. Sagawa et al. [2019] and Sagawa et al. [2020] introduced worst-group accuracy as a robustness metric to evaluate models under spurious feature reliance. More recently, Kirichenko et al. [2022] and Qiu et al. [2024] analyzed how training dynamics and data complexity affect the emergence of spurious features. Beyond metrics, algorithmic approaches include GroupDRO, which directly minimizes worst-group risk by adaptively reweighting groups during training and thus requires group annotations [Sagawa et al., 2019]. Complementarily, Just Train Twice (JTT) improves group robustness without group labels via a two-stage procedure—train a standard ERM model, mark misclassified examples as proxies for minority groups, and retrain with upweighting [Liu et al., 2021a]. Both methods are supervised and label-driven, whereas our approach regularizes representations during self-supervised pretraining. For certified robustness, Mustafa et al. [2024] prove non-vacuous adversarial population-risk bounds via randomized smoothing and PAC-Bayes for stochastic networks—complementary to our SSRL regularization.

**Self-Supervised Learning and Shortcut Representations**    Recent work has shown that self-supervised representation learning (SSRL), particularly contrastive methods, is also susceptible to spurious features [Hamidieh et al., 2024, Zhu et al., 2023, Ye et al., 2023]. While contrastive learning is a principled route to task-agnostic representations—supported by strong empirical results and theory giving generalization guarantees under both i.i.d. and non-i.i.d. sampling regimes, as well as in adversarial settings with logarithmic negative-sample dependence [Chen et al., 2020, He et al., 2020, Grill et al., 2020, Chen and He, 2021, Caron et al., 2020, Zbontar et al., 2021, Hieu et al., 2025, Hieu and Ledent, 2025, Ghanooni et al., 2024]—it can nevertheless emphasize dominant directions in feature space, encoding shortcuts rather than robust, transferable signals. Empirically, the effects of dominant, easy-to-learn features have been widely documented [Liu et al., 2021b, Jiang et al., 2021a,b, Chen et al., 2021], and studies addressing group robustness or fairness in SSRL often rely on group information or labeled data [Song et al., 2019, Tsai et al., 2020, Wang et al., 2021, Bordes et al., 2022, Scalbert et al., 2023]. In contrast, Robinson et al. [2021] tackled shortcut learning in contrastive setups by employing adversarial feature modification without requiring group labels.

**Spectral Structure and Representation Quality**    The spectral properties of learned representations have been studied in various contexts [He et al., 2024, Bansal et al., 2018, Jing et al., 2021]. Rahaman et al. [2019] and Kalimeris et al. [2019] analyzed how gradient-based optimization favors low-frequency, high-energy components early in training. Xue et al. [2023] examined the role of eigenspectrum bias in the evolution of feature learning. Our work builds on these insights by showing that spectral imbalance—an overconcentration of variance in a few directions—can amplify spurious features and degrade generalization.

**Spectral Regularizers and Low-Rank Perspectives**    In representation learning, several recent methods explicitly incorporate regularizers to encourage more diverse, high-rank representations in self-supervised learning. For example, Barlow Twins [Zbontar et al., 2021] introduces a redundancy-reduction loss that drives the cross-correlation matrix of twin network embeddings toward the identity, thereby decorrelating feature dimensions and minimizing redundancy. Similarly, VICReg adds a variance preservation term and a covariance penalty to maintain per-dimension variance while pushing off-diagonal covariances toward zero, thus preventing informational collapse by decorrelating features [Bardes et al., 2021]. Whitening-based approaches go even further by forcing the entire feature covariance to match an identity matrix (full whitening), which is equivalent to enforcing a full-rank embedding space and comes with theoretical guarantees against dimensional collapse [Ermolov et al., 2021]. Beyond these objectives on the representation statistics, other techniques like Implicit Feature Modification (IFM) actively perturb training examples in feature space to ensure the encoder utilizes a broader set of features, reducing reliance on any single dominant "shortcut" and promoting greater feature diversity [Robinson et al., 2021]. Complementing these regularizers, prior work has explored how low-rank structures relate to generalization. Sainath et al. [2013] found that low-rank projections can improve speech recognition models, and Feng et al. [2022] showed that adversarially trained networks tend to learn flatter spectra. However, Andriushchenko et al. [2023] demonstrated that flatness alone does not guarantee robustness. All of these strategies underline the importance

of promoting feature diversity and high effective dimensionality in learned representations. Our approach differs by emphasizing a balanced eigenspectrum (neither overly flat nor too concentrated) to mitigate spurious feature dominance without the extremes of full whitening or low-rank collapse.

# 4 Theory

This section analyzes how the spectral structure of learned representations affects generalization in downstream tasks. We show that the downstream generalization error depends critically on the spectrum of the feature covariance matrix. Finally, we prove that representations with a uniform eigenspectrum minimize expected generalization error over random downstream tasks, providing a theoretical justification for promoting spectral diversity.

## 4.1 Problem Setup

We consider a learning setup where the inputs $\mathbf{x} \in \mathcal{X} \subset \mathbb{R}^{d'}$ are composed of core and spurious features. Only the core features are causally relevant to the ground-truth labels, but spurious features may correlate with labels due to dataset biases.

We assume access to unlabeled data in the form of similar pairs $(\mathbf{x}, \mathbf{x}^+)$ and independent negatives $\mathbf{x}^-$. Learning proceeds over a function class $\mathcal{F} = \{f : \mathcal{X} \to \mathbb{R}^d \mid \|f(\mathbf{x})\|_2 \leq R\}$, where $f$ maps inputs to $d$-dimensional representations. During training, we optimize an encoder $f \in \mathcal{F}$ jointly with a projection head $h(\mathbf{x}) = W_2 \sigma(W_1 f(\mathbf{x}))$, where $W_1 \in \mathbb{R}^{d_h \times d}$, $W_2 \in \mathbb{R}^{d_p \times d_h}$, and $\sigma(\cdot)$ is a nonlinearity (e.g., ReLU). The encoder and projection head are trained using contrastive losses such as InfoNCE [Oord et al., 2018]. After training, the projection head $h$ is discarded, and the learned features are represented by the matrix

$$F = [f(\mathbf{x}_1), \ldots, f(\mathbf{x}_n)]^\top \in \mathbb{R}^{n \times d},$$

where $n$ is the number of samples.

To evaluate the quality of the learned representations, we freeze the encoder $f$ and train a linear classifier $g_{\mathbf{w}}(f(\mathbf{x})) = \langle f(\mathbf{x}), \mathbf{w} \rangle$ with $\mathbf{w} \in \mathbb{R}^d$. Given a labeled dataset $S = \{(\mathbf{x}_i, y_i)\}_{i=1}^n$ drawn i.i.d. from a distribution $\mathcal{D}$ over $\mathcal{X} \times \mathcal{Y}$, where $\mathbf{x}_i \in \mathbb{R}^{d'}$ and $y_i \in \{-1, +1\}$. The classification downstream task involves minimizing the supervised loss

$$\ell(y, g_{\mathbf{w}}(f(\mathbf{x}))) = (1 - y g_{\mathbf{w}} f(\mathbf{x}))^2.$$

This setup captures the realistic setting where similarity is defined semantically but is not perfectly aligned with task-relevant (core) features, allowing spurious correlations to arise during representation learning.

## 4.2 The Role of Feature Spectra in Downstream Task Generalization

We now investigate how the spectral structure of the feature matrix influences the generalization performance of downstream tasks. First, we analyze the impact of the feature matrix $F$ on the generalization of linear classifiers trained via stochastic gradient descent (SGD). The following corollary, based on Arora et al. [2019], summarizes this effect for a fixed downstream task.

**Corollary 3** (Linear classifier generalization bound under rank-deficient feature matrix). *Let $S = \{(\mathbf{x}_i, y_i)\}_{i=1}^n$ be drawn i.i.d. from a distribution over inputs and binary labels $y_i \in \{\pm 1\}$, and fix a failure probability $\delta \in (0, 1)$. Let $f(\mathbf{x}) \in \mathbb{R}^d$ be a fixed representation and define the feature matrix $F \in \mathbb{R}^{n \times d}$ with rows $f(\mathbf{x}_i)^\top$. Consider a linear predictor $g_{\mathbf{w}}(\mathbf{x}) = \langle f(\mathbf{x}), \mathbf{w} \rangle$ trained with gradient descent. Then, with probability at least $1 - \delta$, the population loss $L_{\mathcal{D}}(g_{\mathbf{w}^{(k)}}) := \mathbb{E}_{(\mathbf{x}, y) \sim \mathcal{D}}[\ell(g_{\mathbf{w}}(\mathbf{x}), y)]$ satisfies*

$$L_{\mathcal{D}}(g_{\mathbf{w}^{(k)}}) \leq \widetilde{O}\left(\sqrt{\frac{\mathbf{y}^\top (FF^\top)^+ \mathbf{y}}{n}}\right),$$

*where $\mathbf{y} = (y_1, \ldots, y_n)^\top$ and $(\cdot)^+$ denotes the Moore–Penrose pseudoinverse. Equivalently, if $FF^\top = \sum_{i=1}^r \lambda_i \mathbf{v}_i \mathbf{v}_i^\top$ with $\lambda_i > 0$ and orthonormal $\{\mathbf{v}_i\}_{i=1}^r$, then*

$$\mathbf{y}^\top (FF^\top)^+ \mathbf{y} = \sum_{i=1}^r \frac{1}{\lambda_i} (\mathbf{v}_i^\top \mathbf{y})^2.$$

*Here $\widetilde{O}$ hides logarithmic factors and dependence on $\delta$.*

The formal statement and proof are provided in Appendix B. The dominant term, $\mathbf{y}^\top (FF^\top)^+ \mathbf{y}$, shows that generalization improves when the label vector $\mathbf{y}$ aligns well with the top eigenspaces of $FF^\top$. In contrastive learning, however, the downstream task is not known during pretraining, so it is unclear which directions in the feature space will ultimately be important.

To address this, we consider downstream tasks that arise by randomly sampling two latent classes $c^+, c^- \in \mathcal{C}$ according to a distribution $\rho$. For each such pair, we assume the existence of class-specific vectors $\mathbf{v}_{c^+}, \mathbf{v}_{c^-}$ such that the optimal linear classifier in the feature space is given by $\mathbf{v} = \mathbf{v}_{c^+} - \mathbf{v}_{c^-}$. Specifically, the class posterior is given by

$$\mathbb{P}(Y_i = +1 \mid \mathbf{v}) = \frac{1 + (F\mathbf{v})_i}{2}, \quad \mathbb{P}(Y_i = -1 \mid \mathbf{v}) = \frac{1 - (F\mathbf{v})_i}{2},$$

where $F \in \mathbb{R}^{n \times d}$ is the feature matrix.

Since downstream tasks are unknown at pretraining time, designing robust representations for contrastive learning requires optimizing for generalization over a distribution of tasks. Assuming $\rho$ is uniform over class pairs, we study which spectral properties of $FF^\top$ lead to improved average generalization. Specifically, we aim to minimize the expected surrogate loss:

$$\mathcal{L}(F) := \mathbb{E}_{\mathbf{v},Y} \left[ Y^\top (FF^\top)^+ Y \right],$$

where the expectation is over random task vectors $\mathbf{v}$ and induced labels $Y \in \{\pm 1\}^n$.

The following theorem shows the optimal structure of $F$ to enhance the generalization on a general downstream task.

**Theorem 4** (Optimal structure under trace constraint (informal)). *Let $F \in \mathbb{R}^{n \times d}$ be a feature matrix with $\mathrm{rank}(F) = r \leq \min(n, d)$, and let $G := FF^\top \in \mathbb{R}^{n \times n}$. Suppose $G$ has eigenvalues $\lambda_1 \geq \lambda_2 \geq \cdots \geq \lambda_r > 0$, with $\lambda_{r+1} = \cdots = \lambda_n = 0$, and a fixed trace constraint $\sum_{i=1}^r \lambda_i = c$. Then the expected loss $\mathcal{L}(F)$ under the random task model is minimized when $FF^\top$ has a uniform spectrum; that is, all non-zero eigenvalues are equal: $\lambda_1 = \cdots = \lambda_r$.*

**Remark 5.** *The trace constraint used in Theorem 4 is both theoretically meaningful and practically justified. In modern contrastive learning pipelines—such as SimCLR and SimSiam—it is standard to apply $\ell_2$-normalization to the feature vectors. This ensures that the overall energy of the representation, captured by $\mathrm{tr}(FF^\top)$, remains approximately constant across batches. Thus, the fixed-trace constraint reflects common empirical practice. Furthermore, since downstream classifiers are scale-invariant, normalizing the trace removes trivial rescalings of the feature matrix and makes the spectral analysis of feature representations well-posed.*

The formal statement and proof are provided in Appendix C. Theorem 4 shows that, when the downstream task is unknown, the optimal feature geometry (under a fixed-trace constraint) is attained when $G := FF^\top$ has a *uniform* eigenspectrum. In the rank-deficient setting this means the eigenvalues are constant on the rank-$r$ support and zero elsewhere. Geometrically, the representation spreads variance evenly across the task-relevant subspace while suppressing degenerate directions, preventing any single mode from dominating. See Appendix I for an illustrative example with spurious correlations that further motivates the uniform spectrum condition.

# 5 Algorithms

We now present a simple and effective regularization method to promote spectral diversity during contrastive learning. To justify this approach, we establish a spectral relationship between the singular values of the feature matrix $F$ and the eigenvalues of its covariance feature matrix $FF^\top$.

## 5.1 Increasing the Rank of Feature Matrix with Spectral Regularization

In a given neural network model, let $F$ denote the feature matrix at the last layer $L$, which corresponds to the output of the encoder. Our objective is to flatten the spectrum of the matrix $FF^\top$ during the pretraining stage, thus encouraging the model to equalize the eigenvalues and, as a result, increase the effective rank of the feature matrix. To achieve this, we propose a regularizer that efficiently facilitates this objective:

Table 1: Worst-group accuracy (%) for SSRL methods (SimCLR, SimSiam), SimCLR-LateTVG, SimSiam-LateTVG, and our method, which in this experiment is applied only to SimCLR. Values are reported as mean ± standard deviation across 5 random seeds for our method. Results for other methods are taken from Hamidieh et al. [2024].

| DATASET | SIMCLR | SIMSIAM | SIMCLR-LATETVG | SIMSIAM-LATETVG | OURS |
|---|---|---|---|---|---|
| CMNIST | 81.7 | 80.7 | 83.8 | 83.1 | **95.10 ± 2.90** |
| SPURCIFAR-10 | 36.5 | 43.4 | 40.4 | 61.4 | **64.65 ± 2.86** |
| CELEBA | 76.7 | 77.5 | 82.2 | 83.1 | **83.96 ± 1.65** |
| METASHIFT | 45.5 | 42.3 | 59.3 | **79.6** | 61.14 ± 8.02 |
| WATERBIRDS | 43.8 | 48.3 | 55.4 | **56.3** | 50.25 ± 0.79 |

Table 2: Average accuracy (%) for SSRL methods (SimCLR, SimSiam), SimCLR-LateTVG, SimSiam-LateTVG, and our method, which in this experiment is applied only to SimCLR. Values are reported as mean ± standard deviation across 5 random seeds for our method. Results for other methods are taken from Hamidieh et al. [2024].

| DATASET | SIMCLR | SIMSIAM | SIMCLR-LATETVG | SIMSIAM-LATETVG | OURS |
|---|---|---|---|---|---|
| CMNIST | 82.5 | 82.1 | - | 80.6 | **98.08 ± 0.26** |
| SPURCIFAR-10 | 69.3 | 75.1 | - | 76.1 | **82.85 ± 0.82** |
| CELEBA | 82.1 | 81.9 | - | **88.9** | 88.40 ± 0.35 |
| METASHIFT | 55.1 | 55.8 | - | 70.1 | **76.57 ± 1.01** |
| WATERBIRDS | 47.5 | 50.7 | - | 54.8 | **58.32 ± 0.59** |

**Definition 6** (Spectrum Flattening Regularizer). *Let $G := FF^\top$ have eigenvalues $\lambda_1, \ldots, \lambda_r > 0$ (with $r = \mathrm{rank}(G)$) and set $p_i := \lambda_i / \sum_{j=1}^{r} \lambda_j$. Define*

$$\mathcal{R}_{spec}(G) \;=\; \sum_{i=1}^{r} p_i^2.$$

*This penalizes spectral concentration and is minimized at the uniform spectrum ($p_i = 1/r$).*

To encourage more uniformly distributed representations, we incorporate the spectrum flattening regularizer into the contrastive learning objective. The resulting loss function is defined as

$$\mathcal{L} = \mathcal{L}_{\text{SSRL}} + \alpha \mathcal{R}_{\text{spec}}(FF^\top), \tag{1}$$

where $\mathcal{L}_{\text{SSRL}}$ denotes the standard contrastive loss (e.g., SimCLR), and $\alpha \in \mathbb{R}_+$ is a weighting coefficient that controls the influence of the regularization term. The addition of $\mathcal{R}_{\text{spec}}$ encourages the learned feature representations to maintain a flatter spectral profile, thereby promoting greater feature diversity and mitigating representational collapse. Pseudocode for computing the regularization term is provided in Appendix E.

## 6 Experiments

We evaluate our method by analyzing how flattening the spectrum of the feature matrix improves worst-group accuracy under spurious correlations. Motivated by Lemma 4 in Appendix D, we apply the regularizer directly to the (centered) mini-batch feature Gram matrix and add it to the SimCLR objective to promote robust feature learning without labels or group annotations. Concretely, for each mini-batch $B$ with centered features $\tilde{F}_B \in \mathbb{R}^{|B| \times d}$, let $G_B := \tilde{F}_B \tilde{F}_B^\top$ have nonzero eigenvalues $\{\lambda_i\}_{i=1}^{r_B}$ and define $p_i := \lambda_i / \sum_{j=1}^{r_B} \lambda_j$. The spectrum–flattening regularizer is

$$\mathcal{R}_{\text{spec}}(B) \;=\; \sum_{i=1}^{r_B} p_i^2,$$

which penalizes concentration of energy and is minimized at a uniform spectrum. (Equivalently, the "zero-at-optimum" form $r_B \sum_{i=1}^{r_B} (p_i - \frac{1}{r_B})^2$ can be used.) Additional details are provided in Appendix E.

## 6.1 Baseline Methods

We compare our approach with SSRL models pretrained using standard SSRL losses, specifically SimCLR [Chen et al., 2020] and SimSiam [Chen and He, 2021]. SimCLR employs a contrastive learning framework, where paired augmented views of the same image are pulled together in the feature space while different images are pushed apart using a contrastive loss. It consists of an encoder (e.g., ResNet), a projection head, and a contrastive loss function that requires negative samples. In contrast, SimSiam is a non-contrastive method that avoids negative samples by leveraging a stop-gradient mechanism and a predictor network to prevent collapsed representations. Additionally, we evaluate our method against LateTVG [Hamidieh et al., 2024], a state-of-the-art SSRL approach designed to mitigate spurious correlations during pretraining, which has been applied to both SimCLR and SimSiam. We include baseline results for Barlow Twins [Zbontar et al., 2021], DirectDLR [Jing et al., 2021], IFM [Robinson et al., 2021], BYOL [Grill et al., 2020], and DINO [Caron et al., 2021] in Appendix H for completeness and comparison.

## 6.2 Datasets

We evaluate all methods on five widely used vision benchmarks designed to study spurious correlations. Among them, SpurCIFAR-10 [Nagarajan et al., 2020] and C-MNIST [Arjovsky et al., 2019] are synthetic datasets constructed by introducing strong artificial correlations: SpurCIFAR-10 modifies CIFAR-10 images by associating the color of horizontal lines with the object class, with a spurious correlation strength of 0.95, while CMNIST colors the MNIST digits to create a spurious correlation between digit color and binary label, with a spurious correlation strength of 0.99. The remaining datasets are based on real-world imagery. In CelebA [Liu et al., 2015], gender (female or male) is spuriously correlated with hair color (blond or non-blond). MetaShift [Liang and Zou, 2022] explores spurious correlations in the Cats and Dogs classes, where the background (indoor or outdoor) is associated with the type of pet (cat or dog). Finally, Waterbirds [Sagawa et al., 2019] contains a spurious correlation between the background (land or water) and the bird species (landbird or waterbird). Together, these benchmarks cover both synthetic and natural settings, allowing us to evaluate robustness across different types and strengths of spurious correlations.

## 6.3 Implementation Details

**SSRL Pretraining**    For each dataset, we train an encoder (ResNet-18 or ResNet-50) with a projection head composed of two linear layers separated by a ReLU activation. The training follows the contrastive learning framework of SimCLR, using the contrastive loss as the primary objective. Additionally, we incorporate our regularizer from Equation 1 to enhance feature representations. To determine the optimal regularization weight ($\alpha$), we perform a grid search over the values 0.001, 0.005, 0.01, 0.05, 0.1. The regularizer flattens the spectrum of the feature matrix, effectively increasing its rank by applying the regularization within each batch. This design ensures seamless integration with the SSRL loss function while preventing the rank from increasing excessively, which could lead to the learning of irrelevant features.

**Downstream Task**    After pretraining, we freeze the encoder and use its output representations for linear probing. For linear probing, a linear classifier is trained on top of these fixed representations using cross-entropy loss, with only the classifier weights being updated. To ensure a balanced training dataset, we subsample the majority groups [Sagawa et al., 2020, Idrissi et al., 2022], helping to mitigate geometric biases in the linear classifier [Nagarajan et al., 2020]. Finally, we evaluate the learned representations on the standard test split of each dataset, leveraging group information to report both average accuracy and worst-group accuracy.

## 6.4 Results

Our method achieves consistently higher worst-group and average accuracies compared to strong SSRL baselines, demonstrating its robustness across diverse datasets without relying on architectural

modifications. Table 1 reports worst-group accuracies across all five datasets, averaged over five random seeds, while Table 2 shows the corresponding average accuracies. Results for baselines SimCLR and SimSiam are taken from Hamidieh et al. [2024].

Our approach consistently outperforms LateTVG when combined with SimCLR across four datasets (except for waterbirds dataset) and both evaluation metrics. When paired with SimSiam, it remains highly competitive—matching or exceeding LateTVG on several benchmarks—while retaining a simpler architecture. Notably, our SimCLR experiments use 512-dimensional embeddings, whereas LateTVG's SimSiam results are based on 2048-dimensional representations. We have also applied our regularizer to SimSiam; the complete results appear in Appendix H. We further observe that Hamidieh et al. [2024] does not provide the pruning parameter that determines their reported results, nor does it report the standard deviations of the accuracies. Worst-group accuracy can fluctuate substantially, so the absence of variance estimates prevents a rigorous assessment of the consistency of their reported gains.

Moreover, while LateTVG [Hamidieh et al., 2024] is tailored for architectures with stop-gradient operations, our approach is architecture-agnostic and achieves comparable or superior performance with a simpler 512-dimensional feature space. This highlights the potential of spectral regularization for broad SSRL applications. Hyperparameter settings for our method are provided in Appendix F, and additional ablation studies are included in Appendices G and H.

### 6.5 Sensitivity to Regularization Strength

We evaluate the sensitivity of the spectral regularization strength $\alpha_{\text{spec}}$ on MetaShift with SimCLR, keeping all other settings and the random seed fixed. The spectral regularizer is added to the SimCLR objective (as defined earlier). Table 3 shows that performance peaks at $\alpha_{\text{spec}}=0.001$ (**75.84%**), dips as $\alpha_{\text{spec}}$ increases to 0.010 (66.29%), partially recovers at 0.025 (73.60%), and degrades again for $\geq 0.050$ (66–68%). These results indicate sensitivity: small regularization works best, while stronger penalties lead to underperformance; a practical starting point is $\alpha_{\text{spec}} \in [0.001, 0.005]$.

Table 3: Sensitivity of average accuracy (%) on MetaShift to spectral regularization strength $\alpha_{\text{spec}}$ with SimCLR (fixed seeds). Best result are in **bold**.

| $\alpha_{\text{spec}}$ | 0.001 | 0.0025 | 0.005 | 0.010 | 0.025 | 0.050 | 0.100 |
|---|---|---|---|---|---|---|---|
| Avg. Accuracy (%) | 75.84 | 71.91 | 68.54 | 66.29 | 73.60 | 67.42 | 66.29 |

## 7 Conclusion

In this paper, we addressed the challenge of spurious feature reliance in self-supervised representation learning by providing both theoretical insights and a practical solution. We showed that the expressiveness of learned representations is closely tied to the number of significant singular values of the feature matrix, which in turn impacts downstream task performance. To promote richer and more diverse representations, we introduced a spectrum-flattening regularizer that increases the effective rank of the feature space. Rather than explicitly removing spurious features, our method encourages the learning of a broader set of features beyond spurious correlations, thereby improving generalization to downstream tasks without requiring labels or group annotations.

## Acknowledgments and Disclosure of Funding

Part of this work was conducted within the DFG Research Unit FOR 5359 on Deep Learning on Sparse Chemical Process Data (BU 4042/2-1, KL 2698/6-1, and KL 2698/7-1). MK and SF further acknowledge support by the DFG through TRR 375 (ID 511263698), SPP 2298 (KL 2698/5-2), and SPP 2331 (FE 2282/1-2, FE 2282/6-1, and KL 2698/11-1), by the Carl-Zeiss Foundation through the initiatives AI-Care, AI4ChemRisk, and Process Engineering 4.0, and by the BMFTR award 01IS24071A.

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

Figure 3: Effect of spectral manipulations on SimCLR-trained feature representations for SpurCIFAR-10. We evaluate trained features with a linear classifier where singular values are incrementally truncated (from 0 to 512) either with (3a) or without (3b) flattening the remaining spectrum. Flattening alone (even with no truncation) significantly improves worst-group accuracy—from 30% to 40%—highlighting the importance of spectral balance for robust representation learning. Shaded bands indicate worst- and best-group accuracies.

# A    Ablation Study on Removing Small Singular Values Without Flattening the Spectrum

To test our hypothesis, we manipulate the spectrum of the feature matrix learned by SimCLR on SpurCIFAR-10. Specifically, we compare two interventions: (1) progressively truncating the smallest singular values while flattening the remaining spectrum (Figure 3a; the same as Figure 2b in the main paper), and (2) progressively truncating the smallest singular values without flattening the remaining spectrum (Figure 3b). We then evaluate the quality of the resulting representations using a linear classifier.

While both approaches yield some gains in worst-group accuracy as more low-variance directions are removed, the second approach (truncation without flattening) exhibits instability (see Figure 3b); it is unclear which singular values should be removed to consistently improve performance. In contrast, Figure 3a shows that truncation combined with flattening leads to more robust and consistent improvements in worst-group accuracy. Notably, even without any truncation, simply flattening the full spectrum significantly boosts worst-group performance, from approximately 30% to 40%.

Taken together, these findings suggest that low-rank directions provide limited benefit for generalization. As illustrated in Figures 3, a balanced spectrum plays a crucial role in enabling robust representations, reinforcing the importance of spectral regularization in the presence of spurious correlations.

# B    Proof of Corollary 3

**Corollary 3** (Linear classifier generalization bound under rank-deficient feature matrix). *Let $S = \{(\mathbf{x}_i, y_i)\}_{i=1}^n$ be drawn i.i.d. from a distribution over inputs and binary labels $y_i \in \{\pm 1\}$, and fix a failure probability $\delta \in (0, 1)$. Let $f(\mathbf{x}) \in \mathbb{R}^d$ be a fixed representation and define the feature matrix $F \in \mathbb{R}^{n \times d}$ with rows $f(\mathbf{x}_i)^\top$. Consider a linear predictor $g_{\mathbf{w}}(\mathbf{x}) = \langle f(\mathbf{x}), \mathbf{w} \rangle$ trained with gradient descent. Then, with probability at least $1 - \delta$, the population loss $L_{\mathcal{D}}(g_{\mathbf{w}^{(k)}}) := \mathbb{E}_{(\mathbf{x}, y) \sim \mathcal{D}}[\ell(g_{\mathbf{w}}(\mathbf{x}), y)]$ satisfies*

$$L_{\mathcal{D}}(g_{\mathbf{w}^{(k)}}) \leq \widetilde{O}\left( \sqrt{\frac{\mathbf{y}^\top (FF^\top)^+ \mathbf{y}}{n}} \right),$$

*where* $\mathbf{y} = (y_1, \ldots, y_n)^\top$ *and* $(\cdot)^+$ *denotes the Moore–Penrose pseudoinverse. Equivalently, if* $FF^\top = \sum_{i=1}^r \lambda_i \, \mathbf{v}_i \mathbf{v}_i^\top$ *with* $\lambda_i > 0$ *and orthonormal* $\{\mathbf{v}_i\}_{i=1}^r$, *then*

$$\mathbf{y}^\top (FF^\top)^+ \mathbf{y} = \sum_{i=1}^r \frac{1}{\lambda_i} (\mathbf{v}_i^\top \mathbf{y})^2.$$

*Here* $\widetilde{O}$ *hides logarithmic factors and dependence on* $\delta$.

We now prove the result for a full-rank feature matrix.

*Proof.* Without loss of generality, assume $F$ is appropriately normalized, such that $\lambda_{\max}(FF^T) \le 1$. Consider gradient updates of the form

$$\mathbf{w}(k+1) - \mathbf{w}(k) = -\eta \frac{d\Phi}{d\mathbf{w}} = -\eta F^T(\mathbf{g}(k) - \mathbf{y})$$

with $\eta = \mathcal{O}\left(\frac{1}{2\lambda_{\max}(FF^T)}\right)$ and $\mathbf{w}(0) = 0$. The outputs of the linear network evolve as

$$\mathbf{g}(k+1) - \mathbf{g}(k) = F(\mathbf{w}(k+1) - \mathbf{w}(k)) = -\eta FF^T(\mathbf{g}(k) - \mathbf{y})$$

Thus, the distance of the outputs to the labels evolves as

$$\mathbf{g}(k) - \mathbf{y} = \mathbf{g}(k-1) - \eta FF^T(\mathbf{g}(k-1) - \mathbf{y}) - \mathbf{y} = (\mathbf{I} - \eta FF^T)(\mathbf{g}(k-1) - \mathbf{y}) = (\mathbf{I} - \eta FF^T)^k(\mathbf{g}(0) - \mathbf{y})$$

Using the previous result, we can express the change in the weights during training as

$$\begin{aligned}
\mathbf{w}(K) - \mathbf{w}(0) &= \sum_{k=0}^{K-1} \mathbf{w}(k+1) - \mathbf{w}(k) \\
&= -\sum_{k=0}^{K-1} \eta F^T(\mathbf{g}(k) - \mathbf{y}) \\
&= -\sum_{k=0}^{K-1} \eta F^T(\mathbf{I} - \eta FF^T)^k(\mathbf{g}(0) - \mathbf{y}) \\
&= \sum_{k=0}^{K-1} \eta F^T(\mathbf{I} - \eta FF^T)^k \mathbf{y} - \sum_{k=0}^{K-1} \eta F^T(\mathbf{I} - \eta FF^T)^k \mathbf{g}(0)
\end{aligned}$$

To bound the change in the weights, we bound each term individually.

$$\begin{aligned}
||\eta F^T \sum_{k=0}^{K-1} (\mathbf{I} - \eta FF^T)^k \mathbf{y}||_2^2 &= \mathbf{y}^T \left( \sum_{k=0}^{K-1} (\mathbf{I} - \eta FF^T)^k \right)^T FF^T \left( \sum_{k=0}^{K-1} (\mathbf{I} - \eta FF^T)^k \right) \mathbf{y} \\
&= \mathbf{y}^T \left( \sum_{i=1}^n \frac{1 - (1 - \eta\lambda_i)^K}{\lambda_i} v_i v_i^T \right)^T \sum_{i=1}^n \lambda_i v_i v_i^T \left( \sum_{i=1}^n \frac{1 - (1 - \eta\lambda_i)^K}{\lambda_i} v_i v_i^T \right) \mathbf{y} \\
&= \mathbf{y}^T \left( \sum_{i=1}^n \lambda_i \left( \frac{1 - (1 - \eta\lambda_i)^K}{\lambda_i} \right)^2 v_i v_i^T \right) \mathbf{y} \\
&\le \mathbf{y}^T \left( \sum_{i=1}^n \lambda_i^{-1} v_i v_i^T \right) \mathbf{y} \\
&= \mathbf{y}^T \left( FF^T \right)^{-1} \mathbf{y}
\end{aligned}$$

$$||\sum_{k=0}^{K-1} \eta F^T(\mathbf{I} - \eta FF^T)^k \mathbf{g}(0)||_2 \le \eta\sqrt{n} \left( \sum_{k=0}^{K-1} (1 - \eta\lambda_{\min}(FF^T))^k \right) ||\mathbf{g}(0)||_2 \le 0$$

Taking these bounds together yields

$$\|\mathbf{w}(K) - \mathbf{w}(0)\|_2 \le \sqrt{\mathbf{y}^T \left(FF^T\right)^{-1} \mathbf{y}}$$

Let $\epsilon \in \{\pm 1\}^n$. Then it holds

$$\langle \epsilon, X\mathbf{w} \rangle = \langle \epsilon, X(\mathbf{w} - \mathbf{w}(0)) \rangle + \langle \epsilon, X\mathbf{w}(0) \rangle \le \sqrt{n}\|\mathbf{w} - \mathbf{w}(0)\|_2 + \langle \epsilon, X\mathbf{w}(0) \rangle = \sqrt{n}\|\mathbf{w} - \mathbf{w}(0)\|_2$$

Let $\mathcal{F}_R = \{\langle \mathbf{w}, \mathbf{x} \rangle : \|\mathbf{w} - \mathbf{w}(0)\| \le R\}$.

$$\mathcal{R}_S(\mathcal{F}_R) = \frac{1}{n}\mathbb{E}_{\epsilon \sim \{\pm 1\}^n} \left[ \sup_{\|\mathbf{w} - \mathbf{w}(0)\| < R} \langle \epsilon, X\mathbf{w} \rangle \right] < \frac{1}{\sqrt{n}}R = \frac{1}{\sqrt{n}}R$$

$$\sup_{f \in \mathcal{F}_R} L_D(f) - L_S(f) \le 2\mathcal{R}_S(\mathcal{F}_R) + \mathcal{O}\left(\sqrt{\frac{\log \frac{2}{\delta}}{2n}}\right)$$

$$\le 2\frac{1}{\sqrt{n}}R + \mathcal{O}\left(\sqrt{\frac{\log \frac{2}{\delta}}{2n}}\right)$$

$$= 2\sqrt{\frac{\mathbf{y}^T \left(FF^T\right)^{-1} \mathbf{y}}{n}} + \mathcal{O}\left(\sqrt{\frac{\log \frac{2}{\delta}}{2n}}\right)$$

Using

$$\|\mathbf{g} - \mathbf{y}\|_2 \le \|(\mathbf{I} - \eta FF^T)^k\|_2 \|\mathbf{g}(0) - \mathbf{y}\|_2 \le (1 - \eta\lambda_{\min}(FF^T))^k \sqrt{n} \le 1$$

for sufficiently large $k \ge \log \sqrt{n}$, we can bound $L_S$ as follows.

$$L_S(f) = \frac{1}{n}\sum_{i=1}^{n} |g(f(x_i)) - y|^2 = \frac{1}{n}\|\mathbf{g} - \mathbf{y}\|_2^2 \le \frac{1}{\sqrt{n}}$$

$\square$

We now prove the case in which the feature matrix is rank-deficient.

*Proof.* Without loss of generality, assume $F$ is appropriately normalized so that $\lambda_{\max}(FF^\top) \le 1$. Consider gradient updates of the form

$$\mathbf{w}(k{+}1) - \mathbf{w}(k) = -\eta \frac{d\Phi}{d\mathbf{w}} = -\eta F^\top (\mathbf{g}(k) - \mathbf{y})$$

with $\eta = \mathcal{O}\left(\frac{1}{2\lambda_{\max}(FF^\top)}\right)$ and $\mathbf{w}(0) = 0$. The outputs evolve as

$$\mathbf{g}(k{+}1) - \mathbf{g}(k) = F(\mathbf{w}(k{+}1) - \mathbf{w}(k)) = -\eta FF^\top (\mathbf{g}(k) - \mathbf{y}),$$

hence

$$\mathbf{g}(k) - \mathbf{y} = (\mathbf{I} - \eta FF^\top)^k (\mathbf{g}(0) - \mathbf{y}).$$

Using this, the weight displacement after $K$ steps is

$$\mathbf{w}(K) - \mathbf{w}(0) = -\sum_{k=0}^{K-1} \eta F^\top (\mathbf{g}(k) - \mathbf{y}) = -\sum_{k=0}^{K-1} \eta F^\top (\mathbf{I} - \eta FF^\top)^k (\mathbf{g}(0) - \mathbf{y})$$

$$= \sum_{k=0}^{K-1} \eta F^\top (\mathbf{I} - \eta FF^\top)^k \mathbf{y} - \sum_{k=0}^{K-1} \eta F^\top (\mathbf{I} - \eta FF^\top)^k \mathbf{g}(0).$$

We bound the two terms separately. For the first,

$$\left\| \eta F^\top \sum_{k=0}^{K-1} (\mathbf{I} - \eta \, F F^\top)^k \mathbf{y} \right\|_2^2 = \mathbf{y}^\top \Big( \sum_{k=0}^{K-1} (\mathbf{I} - \eta \, F F^\top)^k \Big)^\top F F^\top \Big( \sum_{k=0}^{K-1} (\mathbf{I} - \eta \, F F^\top)^k \Big) \mathbf{y}.$$

Let $F F^\top = \sum_{i=1}^r \lambda_i \mathbf{v}_i \mathbf{v}_i^\top$ with $\lambda_i > 0$ (the remaining eigenvalues are 0; their contributions vanish in the display above). Then

$$\sum_{k=0}^{K-1} (\mathbf{I} - \eta \, F F^\top)^k = \sum_{i=1}^r \frac{1 - (1 - \eta \lambda_i)^K}{\eta \lambda_i} \, \mathbf{v}_i \mathbf{v}_i^\top,$$

and

$$\mathbf{y}^\top \Big( \sum_k (\cdot) \Big)^\top F F^\top \Big( \sum_k (\cdot) \Big) \mathbf{y}$$
$$= \mathbf{y}^\top \left( \sum_{i=1}^r \lambda_i \left( \frac{1 - (1 - \eta \lambda_i)^K}{\lambda_i} \right)^2 \mathbf{v}_i \mathbf{v}_i^\top \right) \mathbf{y} \; \leq \; \mathbf{y}^\top \left( \sum_{i=1}^r \lambda_i^{-1} \mathbf{v}_i \mathbf{v}_i^\top \right) \mathbf{y} = \mathbf{y}^\top (F F^\top)^+ \mathbf{y},$$

where $(\cdot)^+$ denotes the Moore–Penrose pseudoinverse and we used $(1 - (1 - \eta \lambda_i)^K)^2 \leq 1$.

For the second term, since $\mathbf{w}(0) = 0$ we have $\mathbf{g}(0) = F \mathbf{w}(0) = 0$, hence

$$\left\| \sum_{k=0}^{K-1} \eta \, F^\top (\mathbf{I} - \eta \, F F^\top)^k \mathbf{g}(0) \right\|_2 = 0.$$

Combining the two bounds yields

$$\|\mathbf{w}(K) - \mathbf{w}(0)\|_2 \; \leq \; \sqrt{\mathbf{y}^\top (F F^\top)^+ \mathbf{y}}.$$

Let $\epsilon \in \{\pm 1\}^n$. Then

$$\langle \epsilon, X \mathbf{w} \rangle = \langle \epsilon, X(\mathbf{w} - \mathbf{w}(0)) \rangle + \langle \epsilon, X \mathbf{w}(0) \rangle \leq \sqrt{n} \, \|\mathbf{w} - \mathbf{w}(0)\|_2,$$

and for $\mathcal{F}_R = \{\langle \mathbf{w}, \mathbf{x} \rangle : \|\mathbf{w} - \mathbf{w}(0)\| \leq R\}$,

$$\mathcal{R}_S(\mathcal{F}_R) = \frac{1}{n} \mathbb{E}_\epsilon \Big[ \sup_{\|\mathbf{w} - \mathbf{w}(0)\| \leq R} \langle \epsilon, X \mathbf{w} \rangle \Big] \; \leq \; \frac{R}{\sqrt{n}}.$$

Hence, with probability at least $1 - \delta$,

$$\sup_{f \in \mathcal{F}_R} \big( L_D(f) - L_S(f) \big) \; \leq \; 2 \, \mathcal{R}_S(\mathcal{F}_R) + \mathcal{O}\Big( \sqrt{\tfrac{\log(2/\delta)}{2n}} \Big) \; \leq \; \frac{2R}{\sqrt{n}} + \mathcal{O}\Big( \sqrt{\tfrac{\log(2/\delta)}{2n}} \Big),$$

and taking $R = \|\mathbf{w}(K) - \mathbf{w}(0)\|_2 \leq \sqrt{\mathbf{y}^\top (F F^\top)^+ \mathbf{y}}$ gives the stated bound.

Finally, for the empirical term, decompose $\mathbf{y} = \mathbf{y}_{\text{range}} + \mathbf{y}_{\text{null}}$ along the eigenspaces of $F F^\top$, so that $(\mathbf{I} - \eta \, F F^\top)^k \mathbf{y}_{\text{null}} = \mathbf{y}_{\text{null}}$ and

$$\|\mathbf{g} - \mathbf{y}\|_2 = \|(\mathbf{I} - \eta \, F F^\top)^k (\mathbf{g}(0) - \mathbf{y}_{\text{range}}) - \mathbf{y}_{\text{null}}\|_2 \; \leq \; (1 - \eta \, \lambda_{\min}^+)^k \, \|\mathbf{y}_{\text{range}}\|_2 + \|\mathbf{y}_{\text{null}}\|_2,$$

where $\lambda_{\min}^+$ is the smallest positive eigenvalue of $F F^\top$. Choosing $k$ so that $(1 - \eta \, \lambda_{\min}^+)^k \leq 1/\sqrt{n}$ yields

$$L_S(f) \; = \; \frac{1}{n} \|\mathbf{g} - \mathbf{y}\|_2^2 \; \leq \; \frac{1}{n}\Big( 1 + \|\mathbf{y}_{\text{null}}\|_2^2 \Big) \; \leq \; \frac{1}{n}(1 + n),$$

and in particular $L_S(f) \leq 1$ always; when $\mathbf{y}$ lies largely in $\text{range}(F)$ (the learnable subspace), this term is $o(1)$ for large $n$. $\qquad \square$

# C   Proof of Theorem 4

In this section, we formalize the problem setting, restate Theorem 4 and provide its proof. The dominant term, $\mathbf{y}^\top (FF^\top)^{-1}\mathbf{y}$, shows that generalization improves when the label vector $\mathbf{y}$ aligns well with the top eigenspaces of $FF^\top$. In contrastive learning, however, the downstream task is not known during pretraining, so it is unclear which directions in the feature space will ultimately be important.

To address this, we consider downstream tasks that arise by randomly sampling two latent classes $c^+, c^- \in \mathcal{C}$ according to a distribution $\rho$. For each such pair, we assume the existence of class-specific vectors $\mathbf{v}_{c^+}, \mathbf{v}_{c^-}$ such that the optimal linear classifier in the feature space is given by $\mathbf{v} = \mathbf{v}_{c^+} - \mathbf{v}_{c^-}$. Specifically, the class posterior is given by

$$\mathbb{P}(Y_i = +1 \mid \mathbf{v}) = \frac{1 + (F\mathbf{v})_i}{2}, \quad \mathbb{P}(Y_i = -1 \mid \mathbf{v}) = \frac{1 - (F\mathbf{v})_i}{2},$$

where $F \in \mathbb{R}^{n \times d}$ is the feature matrix.

Since downstream tasks are unknown at pretraining time, designing robust representations for contrastive learning requires optimizing for generalization over a distribution of tasks. Assuming $\rho$ is uniform over class pairs, we study which spectral properties of $FF^\top$ lead to improved average generalization. Specifically, we aim to minimize the expected surrogate loss:

$$\mathcal{L}(F) := \mathbb{E}_{\mathbf{v},Y} \left[ Y^\top (FF^\top)^+ Y \right],$$

where the expectation is over random task vectors $\mathbf{v}$ and induced labels $Y \in \{\pm 1\}^n$.

The following theorem shows the optimal structure of $F$ to enhance the generalization on a general downstream task.

**Theorem 4** (Optimality of Uniform Spectrum under Trace Constraint). *Let $F \in \mathbb{R}^{n \times d}$ be a feature matrix with $\mathrm{rank}(F) = r \le \min(n, d)$, and define $G := FF^\top \in \mathbb{R}^{n \times n}$, where $G$ has eigenvalues $\lambda_1 \ge \lambda_2 \dots \lambda_r > 0$ and $\lambda_{r+1} = \cdots = \lambda_n = 0$. Suppose the trace is fixed, i.e., $\mathrm{Tr}(FF^\top) = \sum_{i=1}^{r} \lambda_i = c$ for some constant $c > 0$. Then the expected quadratic form*

$$\mathcal{L}(F) := \mathbb{E}_{Y \sim \mathcal{D}} \left[ Y^\top (FF^\top)^{-1} Y \right]$$

*under random task model is minimized when $FF^\top = \lambda I_n$, i.e., when all non-zero eigenvalues are equal.*

*Proof.* We first simplify the objective. Conditioning on $\mathbf{v}$, the second moment of $Y$ satisfies

$$\mathbb{E}[YY^\top \mid \mathbf{v}] = (F\mathbf{v})(F\mathbf{v})^\top + \mathrm{diag}(1 - (F\mathbf{v})^2).$$

Thus,

$$\mathcal{L}(F) = \mathbb{E}_{\mathbf{v}} \left[ \mathrm{tr}\big((FF^\top)^+ (F\mathbf{v})(F\mathbf{v})^\top\big) + \mathrm{tr}\big((FF^\top)^+ \mathrm{diag}(1 - (F\mathbf{v})^2)\big) \right],$$

where $(\cdot)^+$ denotes the Moore–Penrose pseudoinverse (equal to the inverse in the full-rank case).

With the rank-$r$ SVD $F = U_r \Sigma_r V_r^\top$, we have $FF^\top = U_r \Lambda_r U_r^\top$ with $\Lambda_r = \mathrm{diag}(\lambda_1, \dots, \lambda_r)$ and $\lambda_i = \sigma_i^2 > 0$ (and $\lambda_{r+1} = \cdots = \lambda_n = 0$). Using the identity

$$F^\top (FF^\top)^+ F \; = \; V_r I_r V_r^\top \; = \; P_{\mathrm{row}(F)},$$

we obtain

$$\mathrm{tr}\big((FF^\top)^+ (F\mathbf{v})(F\mathbf{v})^\top\big) = \mathbf{v}^\top F^\top (FF^\top)^+ F\mathbf{v} = \|P_{\mathrm{row}(F)}\mathbf{v}\|_2^2.$$

Under the isotropic random-task model for $\mathbf{v}$, the expectation of this term depends only on the rank $r$ (not on the eigenvalues $\{\lambda_i\}$) and is therefore an additive constant for the minimization.

Expanding the second term gives

$$\mathrm{tr}\big((FF^\top)^+ \mathrm{diag}(1 - (F\mathbf{v})^2)\big) = \mathrm{tr}\big((FF^\top)^+\big) - \mathrm{tr}\big((FF^\top)^+ \mathrm{diag}\left((F\mathbf{v})^2\right)\big).$$

Taking expectation over $\mathbf{v}$ and using the same SVD calculus, the $\mathbf{v}$–dependent parts of the two displays above cancel up to a constant independent of $\{\lambda_i\}$. Hence, up to an additive constant that does not affect the minimizer,

$$\mathcal{L}(F) \; = \; \mathrm{tr}\big((FF^\top)^+\big).$$

Next, we optimize $\mathcal{L}(F)$ under the constraint $\operatorname{tr}(FF^\top) = c$. Let $G := FF^\top$ and let its non-zero eigenvalues be $\lambda_1, \ldots, \lambda_r > 0$ (the rest are zero). Then

$$\mathcal{L}(F) = \operatorname{tr}(G^+) = \sum_{i=1}^{r} \lambda_i^{-1}, \quad \text{and} \quad \operatorname{tr}(G) = \sum_{i=1}^{r} \lambda_i = c.$$

Define the Lagrangian:

$$\mathcal{L}(\lambda_1, \ldots, \lambda_r, \mu) = \sum_{i=1}^{r} \lambda_i^{-1} + \mu \left( \sum_{i=1}^{r} \lambda_i - c \right).$$

Taking partial derivatives with respect to each $\lambda_i$ and setting them to zero:

$$\frac{\partial \mathcal{L}}{\partial \lambda_i} = -\lambda_i^{-2} + \mu = 0 \quad \Rightarrow \quad \lambda_i = \frac{1}{\sqrt{\mu}}, \quad \text{for all } i = 1, \ldots, r.$$

Thus, all non-zero $\lambda_i$ are equal. Substituting into the constraint $\sum_{i=1}^{r} \lambda_i = c$ gives:

$$r \cdot \lambda_i = c \quad \Rightarrow \quad \lambda_i = \frac{c}{r}.$$

$\square$

Theorem 4 suggests that when the downstream task is unknown, learning a feature matrix $FF^\top$ with a uniform spectrum is optimal. See also Appendix I for an illustrative example involving spurious correlations, which further motivates the benefits of a uniform spectrum.

## D  Lemma 4

**Lemma 4.** *Let $F \in \mathbb{R}^{n \times d}$ be a feature matrix with singular values $\sigma_1, \ldots, \sigma_r$, where $r = \min(n, d)$, and let $\lambda_1, \ldots, \lambda_r$ denote the eigenvalues of $FF^\top$. Assume the singular values are normalized so that*

$$\sum_{i=1}^{r} (\sigma_i - 1)^2 \leq \varepsilon \quad \text{for some } \varepsilon \in (0, 1].$$

*Then,*

$$\sum_{i=1}^{r} (\lambda_i - 1)^2 \leq C\varepsilon \quad \text{for a constant } C.$$

*Proof.* Let $F \in \mathbb{R}^{n \times d}$ have rank $r = \min(n, d)$, and let its singular value decomposition be $F = U\Sigma V^\top$, where $\Sigma = \operatorname{diag}(\sigma_1, \ldots, \sigma_r)$ with singular values $\sigma_i > 0$. Then the eigenvalues of $FF^\top \in \mathbb{R}^{n \times n}$ are exactly $\lambda_i = \sigma_i^2$ for $i = 1, \ldots, r$, and $0$ otherwise.

By assumption, the singular values are normalized and satisfy

$$\sum_{i=1}^{r} (\sigma_i - 1)^2 \leq \varepsilon.$$

Our goal is to bound

$$\sum_{i=1}^{r} (\lambda_i - 1)^2.$$

To relate these two expressions, we use the identity:

$$(\sigma_i^2 - 1)^2 = (\sigma_i - 1)^2 (\sigma_i + 1)^2.$$

Since $\sigma_i$ are normalized and $(\sigma_i - 1)^2 \leq \varepsilon$, we have $\sigma_i \in [1 - \sqrt{\varepsilon}, 1 + \sqrt{\varepsilon}]$. Therefore,

$$(\sigma_i + 1)^2 \leq (1 + \sqrt{\varepsilon} + 1)^2 = (2 + \sqrt{\varepsilon})^2 \leq 9 \quad \text{for } \varepsilon \leq 1.$$

Then:

$$(\sigma_i^2 - 1)^2 = (\sigma_i - 1)^2(\sigma_i + 1)^2 \leq 9(\sigma_i - 1)^2.$$

Summing over $i = 1, \ldots, r$, we obtain:

$$\sum_{i=1}^{r}(\lambda_i - 1)^2 = \sum_{i=1}^{r}(\sigma_i^2 - 1)^2 \leq 9\sum_{i=1}^{r}(\sigma_i - 1)^2 \leq 9\varepsilon.$$

$\square$

**Remark 5.** *Lemma 4 shows that flattening the singular values of $F$ implies a corresponding flattening of the eigenvalues of $FF^\top$, leading to a more uniform spectrum.*

**Remark 6.** *Thus, to encourage spectrum uniformity of $FF^\top$, it suffices to regularize the singular values of $F$, which is often simpler and more efficient in contrastive learning frameworks.*

# E   Algorithmic Details of Contrastive Pretraining with Spectral Regularization

This section presents the pseudocode for our contrastive pretraining framework. Algorithm 1 outlines the self-supervised training procedure based on SimCLR with optional spectral regularization. Algorithm 2 describes the computation of the spectrum flattening loss.

---

**Algorithm 1** Self-supervised Contrastive Pretraining with Spectrum Regularization

---

**Input:** Encoder $f_\theta$, projection head $g$, temperature $\tau$, augmentation pipeline $\mathcal{T}$, spectral weight $\alpha_{\text{spec}}$, epochs $N$, batch size $B$
Initialize parameters of $f_\theta$ and $g$
**for** epoch = 1 to $N$ **do**
    **for** each mini-batch $\{x_i\}_{i=1}^{B}$ **do**
        *// Stage 1: Data Augmentation*
        Sample two augmentations $t, t' \sim \mathcal{T}$
        $x_i^1 = t(x_i), x_i^2 = t'(x_i)$ for $i = 1, \ldots, B$

        *// Stage 2: Feature Extraction*
        $z_i^1 = g(f_\theta(x_i^1)), \quad z_i^2 = g(f_\theta(x_i^2))$
        **Post-L2 normalization**: $\tilde{z}_i^1 = z_i^1/\|z_i^1\|_2, \quad \tilde{z}_i^2 = z_i^2/\|z_i^2\|_2$
        Stack all views: $\tilde{\mathcal{Z}} = \{\tilde{z}_1^1, \tilde{z}_1^2, \ldots, \tilde{z}_B^1, \tilde{z}_B^2\} \in \mathbb{R}^{2B \times d}$

        *// Stage 3: Loss Computation*

$$\mathcal{L}_{\text{CL}} = \frac{1}{2B}\sum_{i=1}^{2B} -\log\frac{\exp(\text{sim}(\tilde{z}_i, \tilde{z}_{p(i)})/\tau)}{\sum_{j\neq i}\exp(\text{sim}(\tilde{z}_i, \tilde{z}_j)/\tau)}, \quad \text{sim}(a, b) = a^\top b$$

        **if** spectral regularization is enabled **then**
            $\mathcal{L}_{\text{spec}} \leftarrow \texttt{SpectrumLoss}(\tilde{\mathcal{Z}})$    *// Alg. 2*
        **else**
            $\mathcal{L}_{\text{spec}} \leftarrow 0$
        **end if**
        Total loss: $\mathcal{L} = \mathcal{L}_{\text{CL}} + \alpha_{\text{spec}}\mathcal{L}_{\text{spec}}$

        *// Stage 4: Optimization*
        Update $f_\theta, g$ via gradient step on $\nabla_\theta\mathcal{L}$
    **end for**
**end for**
**Output:** Pretrained encoder $f_\theta$

---

Table 4: Hyperparameter settings and encoder architectures for SimCLR pretraining.

| Dataset | Encoder | Learning Rate | Batch Size | Weight Decay | Epochs | Regularizer $\alpha$ |
|---------|---------|---------------|------------|--------------|--------|----------------------|
| celebA | ResNet-50 | 0.01 | 128 | 1e-4 | 400 | 0.01 |
| cmnist | ResNet-18 | 1e-3 | 128 | 1e-5 | 700 | 0.01 |
| MetaShift | ResNet-18 | 0.05 | 256 | 1e-4 | 700 | 0.01 |
| spurcifar-10 | ResNet-18 | 0.02 | 128 | 5e-4 | 800 | 0.01 |
| waterbirds | ResNet-18 | 0.01 | 64 | 1e-3 | 600 | 0.01 |

---

**Algorithm 2** Spectrum Flattening Loss Computation ($\mathcal{L}_{\text{spec}}$)

---

**Input:** Post-L2 batch features $\tilde{Z} \in \mathbb{R}^{M \times d}$ (here $M{=}2B$)
**Output:** Spectrum loss $\mathcal{L}_{\text{spec}}$

*// Center features (remove batch mean)*
$\bar{z} \leftarrow \frac{1}{M} \sum_{i=1}^{M} \tilde{Z}_{i:} \quad ; \quad Z_c \leftarrow \tilde{Z} - \mathbf{1}\bar{z}^{\top}$

*// Covariance in feature space (angular statistics on the sphere)*
$C \leftarrow \frac{1}{M-1} Z_c^{\top} Z_c \in \mathbb{R}^{d \times d}$

*// Eigenvalues and trace normalization*
Compute eigenvalues $\{\lambda_i\}$ of $C$; let $r = \#\{i : \lambda_i > 0\}$
$p_i \leftarrow \lambda_i / \sum_{j=1}^{r} \lambda_j$ for $i = 1, \ldots, r$

*// Spectrum-flattening objective (minimized at uniform)*

$\mathcal{L}_{\text{spec}} \leftarrow \sum_{i=1}^{r} p_i^2$

**Return:** $\mathcal{L}_{\text{spec}}$

---

# F   Hyperparameters

We employed the SimCLR framework to train ResNet encoders for our approach. To ensure a fair comparison on SimCLR, we adopted the same encoder architectures as those used in Hamidieh et al. [2024], using ResNet-18 for all datasets except CelebA, where ResNet-50 was used. Detailed hyperparameter configurations for SimCLR across all datasets are provided in Table 4. To select the regularization strength $\alpha$ for the spectral flattening loss, we performed a grid search over the values $\{0.001, 0.005, 0.01, 0.05\}$ using validation performance on the worst-group accuracy as the selection criterion. The best-performing value was then fixed for each dataset across all evaluation protocols.

# G   Closing the Gap to Supervised Pretraining

SSRL has demonstrated significant potential in narrowing the performance gap with supervised learning approaches, particularly for general representation learning. Similar to Hamidieh et al. [2024], we utilized a consistent encoder model and varied only the pretraining strategies, ensuring that other variables, such as hyperparameter settings and model selection criteria, remained fixed. Notably, supervised pretraining requires labeled data, whereas SSRL methods do not, reducing the overall annotation cost significantly. While this inherently makes the comparison less direct, the goal of this evaluation is to measure how closely SSRL methods, and specifically our proposed approach, can match or surpass supervised pretraining strategies.

Table 5 compares the SSLR-base (SimCLR or SimSiam) method, our proposed method, and the supervised approach. The results highlight how our method narrows the gap with supervised learning in terms of average accuracy. Additionally, in worst-group accuracy, our approach outperforms both SimCLR/SimSiam and the supervised method on datasets such as CelebA, CMNIST, and Waterbirds.

Table 5: Comparison of our pretraining strategy (SimCLR + spectral regularizer) with supervised models in terms of average and worst-group accuracies (%). Our pretraining strategy achieves comparable performance to supervised models, both in terms of average and worst-group accuracies (%), despite not utilizing any ground-truth labels or group information.

| DATASET | AVERAGE ACCURACY | | | WORST-GROUP ACCURACY | | |
|---|---|---|---|---|---|---|
| | SSRL-BASE | OURS | SUPERVISED | SSRL-BASE | OURS | SUPERVISED |
| CELEBA | 82.1 | 88.40 | **91.9** | 77.5 | **83.96** | 81.7 |
| CMNIST | 82.5 | 98.08 | **98.4** | 81.7 | **95.10** | 94.9 |
| METASHIFT | 55.8 | 76.57 | **89.8** | 45.5 | 61.14 | **83.5** |
| SPURCIFAR-10 | 75.1 | 82.85 | **89.9** | 43.4 | 64.65 | **79.6** |
| WATERBIRDS | 50.7 | 58.32 | **67.9** | 48.3 | **50.25** | 41.1 |

Table 6: Worst-group accuracy (%) comparison between state of the art methods and our spectral regularization applied to SimCLR and SimSiam.

| METHOD | CMNIST | SPURCIFAR-10 | CELEBA | METASHIFT | WATERBIRDS |
|---|---|---|---|---|---|
| BARLOW TWINS | 57.05 | 6.00 | 39.99 | 58.33 | 43.13 |
| DIRECTDLR | 90.32 | 20.58 | 68.68 | 53.84 | 47.32 |
| IFM | **96.77** | 53.23 | - | 53.85 | **50.96** |
| BYOL | 95.82 | 50.67 | - | 41.67 | 46.89 |
| DINO | 56.67 | 11.37 | - | 58.33 | 49.03 |
| SIMSIAM + SPEC (OURS) | 94.40 | 50.10 | 74.17 | **69.23** | 49.08 |
| SIMCLR + SPEC (OURS) | 95.10 | **64.65** | **83.96** | 61.14 | 50.25 |

## H   Comparing with More Baselines

We evaluate the effectiveness of our spectral regularization method by comparing it against two representative baselines: Barlow Twins [Zbontar et al., 2021] and DirectDLR [Jing et al., 2021]. In addition, we examine how our regularizer performs when applied on top of SimSiam. We report both average accuracy and worst-group accuracy across five standard spurious correlation benchmarks as shown in Tables 6 and 7.

## I   Example Motivating Importance of Uniforming the Spectrum

We present a simple example to highlight the role of the eigenspectrum of the feature matrix. Let $F$ be a fixed feature matrix with orthonormal eigenvectors $v^+$, $v^-$, and $v^s$, corresponding to eigenvalues $\lambda^+$, $\lambda^-$, and $\lambda^s$, respectively. Here, $v^+$ and $v^-$ represent class-discriminative directions for labels $+1$ and $-1$, while $v^s$ is a spurious direction with spurious correlation strength $\alpha$.

Specifically, the label generation process is as follows: define $v := \frac{1}{2}(v^+ - v^-)$. For each sample $x_i$, let $f_i = f(x_i)$, and define the perturbed direction:

$$v_i = \begin{cases} v, & \text{with probability } 1 - \alpha, \\ \frac{1}{2}(v + v^s), & \text{with probability } \alpha. \end{cases}$$

The label $y_i \in \{\pm 1\}$ is sampled according to:

$$\mathbb{P}(y_i = +1 \mid f_i) = \frac{1 + f_i^\top v_i}{2}.$$

Let $g_i = g_w(f_i)$, and consider the squared loss:

$$\Phi(\mathbf{g}, \mathbf{y}) = \frac{1}{2} \sum_{i=1}^n (1 - y_i g_i)^2.$$

**Lemma 7.** *The expected gradient flow under the randomness of the labels satisfies:*

$$\mathbb{E}[FF^\top \cdot \nabla_{\mathbf{g}} \Phi] = \left( FF^\top \mathbf{g} - \left[ \left(1 - \frac{\alpha}{2}\right) \cdot \frac{1}{2} \lambda^+ v^+ - \left(1 - \frac{\alpha}{2}\right) \cdot \frac{1}{2} \lambda^- v^- + \frac{\alpha}{2} \lambda^s v^s \right] \right).$$

Table 7: Average accuracy (%) of the state of the art methods and our spectral regularization applied to SimCLR and SimSiam.

| METHOD | CMNIST | SPURCIFAR-10 | CELEBA | METASHIFT | WATERBIRDS |
|---|---|---|---|---|---|
| BARLOW TWINS | 93.10 | 22.00 | 84.61 | 64.60 | 54.67 |
| DIRECTDLR | 96.35 | 50.20 | 78.23 | 75.28 | 53.46 |
| IFM | **98.16** | 76.70 | - | **78.65** | 55.78 |
| BYOL | 96.20 | 73.30 | - | 74.71 | 53.70 |
| DINO | 66.05 | 26.28 | - | 73.60 | 50.60 |
| SIMSIAM + SPEC (OURS) | 97.20 | 71.78 | **89.12** | 77.52 | **60.55** |
| SIMCLR + SPEC (OURS) | 98.08 | **82.85** | 88.40 | 76.57 | 58.32 |

This result shows that even a weak spurious correlation ($\alpha \ll 1$) can dominate the training dynamics if $\lambda^s \gg \lambda^+, \lambda^-$. In contrast, under a flat spectrum (i.e., uniform eigenvalues), the influence of the spurious direction scales linearly with $\alpha$, making the model more robust to such noise.

*Proof.* The first step is to compute the loss with respect to each model output $g_i$ which is given by

$$\frac{d\Phi}{dg_i} = -2y_i(1 - y_i g_i).$$

The sources of randomness are from sampling both $y$ and the random mixing of the spurious feature. By the law of total expectation, the expectation with respect to $y$ and $v$ is given by

$$\mathbb{E}_{v_i|f_i}\left[\mathbb{E}_{y_i|x_i,v_i}\left[\frac{d\Phi}{dg_i}\right]\right]$$

The inner expectation is given by

$$\mathbb{E}_{y_i|f_i,v_i}\left[\frac{d\Phi}{dg_i}\right] = -2\mathbb{E}[y_i - y_i^2 g_i] = 2g_i - 2f_i^\top v_i,$$

since $\mathbb{E}[y_i \mid f_i, v_i] = f_i^\top v_i$ and $y_i^2 = 1$. Further observe

$$\mathbb{E}_{v_i}[x_i^\top v_i] = (1-\alpha)f_i^\top v + \alpha f_i^\top\left(\frac{1}{2}(v + v^s)\right) = \left(1 - \frac{\alpha}{2}\right)f_i^\top v + \frac{\alpha}{2}f_i^\top v^s.$$

Combining the above three equations and letting $\tilde{v} := \left(1 - \frac{\alpha}{2}\right)v + \frac{\alpha}{2}v^s$, we get

$$\mathbb{E}_{v_i,y_i}\left[\frac{d\Phi}{dg_i}\right] = 2g_i - 2f_i^\top\tilde{v}.$$

Stacking across all samples, let $\mathbf{g} = [g_1, \ldots, g_n]^\top$. Then:

$$\nabla_{\mathbf{g}}\Phi = (\mathbf{g} - F\tilde{v}).$$

Applying the data covariance operator $FF^\top$ gives:

$$\mathbb{E}[FF^\top \cdot \nabla_{\mathbf{g}}\Phi] = \left(FF^\top\mathbf{g} - FF^\top F\tilde{v}\right).$$

By the assumption that $v^+$, $v^-$, and $v^s$ are orthonormal eigenvectors of $FF^\top$ with eigenvalues $\lambda^+$, $\lambda^-$, and $\lambda^s$, and $v = \frac{1}{2}(v^+ - v^-)$. Then:

$$F\tilde{v} = \left(1 - \frac{\alpha}{2}\right) \cdot \frac{1}{2}Fv^+ - \left(1 - \frac{\alpha}{2}\right) \cdot \frac{1}{2}Fv^- + \frac{\alpha}{2}Fv^s,$$

and applying $FF^\top$:

$$FF^\top F\tilde{v} = \left(1 - \frac{\alpha}{2}\right) \cdot \frac{1}{2}\lambda^+ v^+ - \left(1 - \frac{\alpha}{2}\right) \cdot \frac{1}{2}\lambda^- v^- + \frac{\alpha}{2}\lambda_s v^s.$$

Substituting this expression concludes the proof:

$$\mathbb{E}[FF^\top \cdot \nabla_{\mathbf{g}}\Phi] = \frac{2}{n}\left(FF^\top\mathbf{g} - \left[\left(1 - \frac{\alpha}{2}\right) \cdot \frac{1}{2}\lambda^+ v^+ - \left(1 - \frac{\alpha}{2}\right) \cdot \frac{1}{2}\lambda^- v^- + \frac{\alpha}{2}\lambda^s v^s\right]\right).$$

$\square$

## J  Computational Efficiency of Regularization Terms

Several self-supervised learning methods aim to mitigate representation collapse and redundancy by decorrelating feature dimensions. **Barlow Twins** minimizes the cross-correlation matrix between two views and enforces it to be close to the identity, effectively promoting invariance while discouraging redundancy Zbontar et al. [2021]. **VICReg** combines an invariance term with variance and covariance regularizers, penalizing off-diagonal entries in the covariance matrix Bardes et al. [2021]. Both methods require computing and differentiating through batch-wise matrices of size $d \times d$, incurring a cost of $O(nd^2)$ to form the matrix, and an additional $O(d^2)$ for computing the regularization loss.

**Whitening-based methods**, such as ZCA whitening Ermolov et al. [2021], go further by requiring not only the covariance matrix $F^\top F \in \mathbb{R}^{d \times d}$ but also its inverse square root, computed via eigendecomposition. This results in a total cost of $O(nd^2 + d^3)$, making them significantly more expensive in high-dimensional settings.

In contrast, our **spectral flattening regularizer** only requires access to the singular values of the feature matrix. These can be obtained by computing the eigenvalues of either $F^\top F \in \mathbb{R}^{d \times d}$ or $FF^\top \in \mathbb{R}^{n \times n}$, or by directly applying SVD to $F \in \mathbb{R}^{n \times d}$. Since all three methods yield the same singular values, one can select the most efficient strategy depending on the dimensions of $F$. Specifically, computing eigenvalues of $F^\top F$ is preferred when $d \ll n$, while $FF^\top$ may be used when $n \ll d$. Direct SVD provides a balanced alternative with cost $O(nd^2)$ when $n \geq d$. This makes our method scalable to large batch sizes and embedding dimensions, while remaining competitive with or cheaper than other decorrelation strategies Shigeto et al. [2023].

All experiments using the spectral flattening regularizer were run on a single NVIDIA A100 GPU with 40 GB memory.

Table 8: Computational complexity of regularization terms for different SSRL methods. Here, $n$ is the batch size and $d$ is the feature dimension.

| METHOD | FORWARD + BACKWARD COST |
|---|---|
| BARLOW TWINS ZBONTAR ET AL. [2021] | $O(nd^2 + d^2)$ |
| VICREG BARDES ET AL. [2021] | $O(nd^2 + d^2)$ |
| WHITENING ERMOLOV ET AL. [2021] | $O(nd^2 + d^3)$ |
| SPECTRAL FLATTENING (OURS) | $O(nd^2)$ |

## K  Random-Task Contrastive Setup (Schematic)

This subsection clarifies the random-task model underlying Section 4.2 and its connection to our main results. Section 4.2 generalizes Corollary 3 (fixed downstream task) to a more realistic *contrastive* pretraining scenario where the downstream task is *unknown* at pretraining time. In our random-task framework: (i) two latent classes $c^+, c^- \in \mathcal{C}$ are sampled, (ii) the classifier vector is $\mathbf{v} = \mathbf{v}_{c^+} - \mathbf{v}_{c^-}$, (iii) labels are generated by the soft classifier $\mathbb{P}(Y_i = \pm 1 \mid \mathbf{v}) = \frac{1 \pm (F\mathbf{v})_i}{2}$, and (iv) the generalization objective becomes $\mathbb{E}_{\mathbf{v}, Y}\left[ Y^\top (FF^\top)^{-1} Y \right]$. Theorem 4 then shows this objective is minimized when $FF^\top$ has a *uniform* spectrum, directly motivating our spectrum-flattening regularizer.

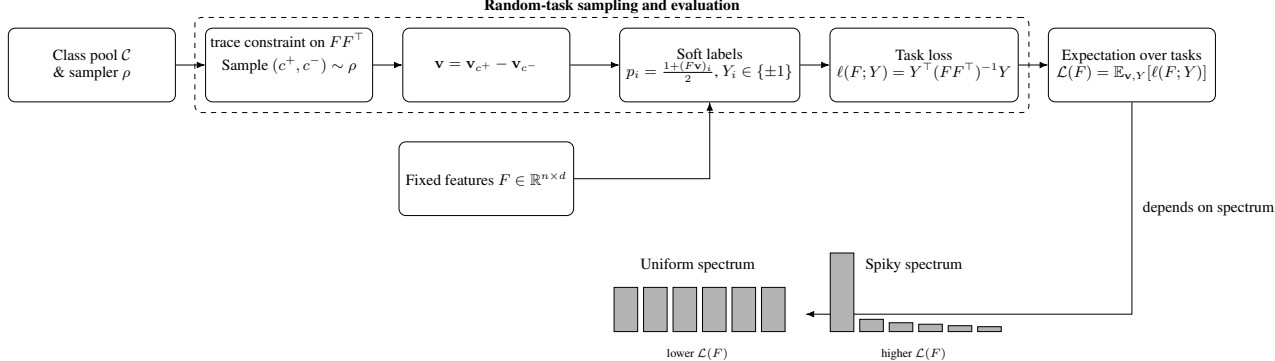

Figure 4: Random-task setup (Section 4.2): sample $(c^+, c^-)$, form $\mathbf{v}$, induce labels via $p_i = \frac{1+(F\mathbf{v})_i}{2}$ and evaluate $\ell(F; Y) = Y^\top (FF^\top)^{-1} Y$. Averaging over tasks yields $\mathcal{L}(F)$, which (under fixed trace) is minimized when $FF^\top$ has a uniform spectrum (Theorem 4).

## L    Limitations

While our spectral flattening regularizer is more efficient than full covariance-based methods in many practical settings, it does incur some additional computational cost due to the need for singular value computation. In our implementation, we compute the singular values of the batch feature matrix $F \in \mathbb{R}^{n \times d}$ via eigendecomposition of $F^\top F$, which scales as $O(nd^2)$ when $n \geq d$. This cost is typically lower than that of Barlow Twins or VICReg, both of which require full covariance matrices and gradients through $d \times d$ terms. However, the quadratic dependence on $d$ may still pose challenges for extremely high-dimensional embeddings. See Appendix J for a detailed comparison of methods and costs.

