# OpenReview forum: "Mitigating Spurious Features in Contrastive Learning with Spectral Regularization"
_NeurIPS.cc/2025/Conference — NeurIPS 2025 poster_

### Official Review · Reviewer_kaxi · 2025-06-26

**Clarity:** 3
**Significance:** 4
**Originality:** 3
**Rating:** 5
**Confidence:** 3

**Summary:**

This paper addresses the problem of spurious feature, also called "simplicity bias", in contrastive learning by examining the eigenvalue spectrum of learned representations. The authors argue that low-rank feature spaces, often a byproduct of learning spurious but easy-to-fit features, limit downstream generalization and robustness, notably on underrepresented data groups.

To address this, the authors introduce a spectrum-flattening regularizer that promotes a more uniform singular value distribution in the feature matrix. This regularization is theoretically motivated by the introduction of a linear classifier generalization bound, and then of a expected inverse energy over random classification tasks which appears to depend on the feature covariance matrix. The authors then showed that the expected inverse energy was minimized whenever the eigenvalues of FF.T are uniform. This simple, architecture-agnostic intervention increases the effective rank of the representation space. Empirical results on SpurCIFAR-10 and other spurious correlation benchmarks demonstrate improvements in worst-group accuracy and overall performance.

**Questions:**

In Theorem 4, you suppose that F satisfies the trace constraint, how valid is this assumption in practice ? Is it trivial ? Could you discuss it a little more ?

In Theorem 4, you argue that when F is a scaled orthogonal matrix, the expected inverse energy is minimized, wouldn't it be relevant to verify in practice that F gets close to orthogonality at the end of the training ?

I found hard to understand the Fig. 2, could it be possible to explain it a bit more a make it clearer at his point of the paper ? You talk about dropping eigenvalues but you regularization do not work this way, so I am having trouble to understand.

**Ethical Concerns:**

["NO or VERY MINOR ethics concerns only"]

**Final Justification:**

Authors have globally responded to my points, I am thus recommending accept.

**Limitations:**

The authors should discuss a bit more their assumptions and compare with more related works as well as other contrastive and non-contrastive methods.

I am prone to changing my opinion to Accept if these limitations are adequately addressed.

**Paper Formatting Concerns:**

No formating concerns.

**Quality:**

3

**Strengths And Weaknesses:**

Strengths
The method is theoretically grounded, offering a clear and intuitive motivation for addressing the spectral collapse caused by spurious features.

The paper is well-written and easy to follow, with a concise experimental setup that effectively communicates the core contributions.

The authors conduct a thoughtful and controlled evaluation, notably by subsampling majority groups to construct balanced training sets, avoiding common pitfalls in spurious correlation benchmarks.

Weaknesses
The paper includes limited comparisons to related work, such as the implicit feature modification (IFM) of Robinson et al., which is cited but not empirically evaluated against.

It lacks broader evaluation across self-supervised learning paradigms. In particular, methods like BYOL and DINO, which exhibit implicit spectral regularization, are not considered—even though relevant theoretical work (e.g., Zhuo et al., ICLR 2023, https://arxiv.org/pdf/2303.02387) highlights spectral properties in non-contrastive frameworks.

The proposed method is not evaluated on standard self-supervised benchmarks such as ImageNet-1k or at least ImageNet-100 or STL-10. Given the method’s robustness to spurious features, one would expect improved generalization performance; such experiments would have helped assess its broader applicability.

---

> ### Author Rebuttal · Authors · 2025-07-31
>
> We sincerely thank the reviewer for the thoughtful summary and constructive feedback. We are especially grateful for recognizing the theoretical grounding of our approach, and the careful design of our experiments. Your comments have been very helpful in guiding our revisions. Below we address your concerns and questions.
>
> **W1: More comparison across SSRL baselines.**
>
> To address the reviewer's concern regarding broader evaluation, we conducted additional experiments with IFM, DINO, and BYOL, beyond the baselines already reported in Table 5 and 6. The results, presented above, show that our spectral regularization maintains competitive or superior worst-group accuracy compared to these methods, especially on SpurCIFAR-10, Waterbirds, and MetaShift. Notably, while IFM performs well on some datasets, our method achieves the highest worst-group accuracy on SpurCIFAR-10 and MetaShift with SimCLR + spec. DINO lags behind on several benchmarks despite its implicit spectral flattening behavior, as noted in prior work.
>
> Table A: Worst-Group Accuracy (%)
> | Dataset      | Barlow Twins | DirectDLR | SimSiam + spec (ours) | SimCLR + spec (ours) | IFM       | BYOL  | DINO      |
> | ------------ | ------------ | --------- | --------------------- | -------------------- | --------- | ----- | --------- |
> | CMNIST       | 57.05        | 90.32     | 94.40                 | 95.10            | **96.77** | 95.82 | 56.67     |
> | SpurCIFAR-10 | 6.00         | 20.58     | 50.10                 | **59.70**            | 53.23     | 50.67 | 11.37     |
> | CelebA       | 39.99        | 68.68     | 74.17                 | **84.20**            | –         | –     | –         |
> | MetaShift    | 58.33        | 53.84     | **69.23**             | 67.40                | 53.85     | 41.67 | 58.33 |
> | Waterbirds   | 43.13        | 47.32     | 49.08                 | **56.70**            | 50.96 | 46.89 | 49.03     |
>
> Table B: Average Accuracy (%)
> | Dataset      | Barlow Twins | DirectDLR | SimSiam + spec (ours) | SimCLR + spec (ours) | IFM       | BYOL      | DINO  |
> | ------------ | ------------ | --------- | --------------------- | -------------------- | --------- | --------- | ----- |
> | CMNIST       | 93.10        | 96.35     | 97.20                 | 97.00                | **98.16** | 96.20     | 66.05 |
> | SpurCIFAR-10 | 22.00        | 50.20     | 71.78                 | **80.10**            | 76.70     | 73.30     | 26.28 |
> | CelebA       | 84.61        | 78.23     | **89.12**             | 88.50                | –         | –         | –     |
> | MetaShift    | 64.60        | 75.28     | 77.52                 | 78.10            | **78.65** | 74.71     | 73.60 |
> | Waterbirds   | 54.67        | 53.46     | 60.55             | 57.90                | 55.78     | 53.70 | 50.60 |
>
> These results demonstrate that our method is not only effective across contrastive methods (SimCLR, SimSiam, IFM) but also compares favorably to non-contrastive approaches like BYOL and DINO. All experiments used a single seed for consistency. This broader evaluation strengthens the case for the general applicability and robustness of our regularizer across different SSRL paradigms.
>
> **W2: More experiments on large-scale datasets: Hard ImageNet [1] and STL-10 [3].**
> We conducted experiments on Hard ImageNet [1]. We report the average accuracy on the standard test split (None) for one random seed:
>
> | Method                     | Avg. Accuracy |
> | ------------------------- | -----------------------------: |
> | SimCLR (baseline)         |                          79.06 |
> | SimSiam (baseline)        |                          79.50 |
> | Ours (SimCLR + Reg.)      |                      79.46 |
> | LATETVG [2] |                          78.00 |
>
> Moreover, we conducted additional experiments on STL-10 using SimCLR as the base method and evaluated the impact of our spectral regularizer under three different regularization strengths. While this was done using a single seed for completeness, the results consistently show improvements in both average and worst-group accuracy over the baseline. These results will be included in the final version of the paper upon acceptance.
>
> | Method               | Regularizer α | Avg. Accuracy (%) | Worst-Group Accuracy (%) |
> | -------------------- | ------------- | ----------------- | ------------------------ |
> | SimCLR (baseline)    | –             | 63.72             | 44.25                    |
> | SimCLR + spec (ours) | 0.001         | 64.20             | 44.75                    |
> | SimCLR + spec (ours) | 0.005         | 64.49             | 45.37                    |
> | SimCLR + spec (ours) | 0.010         | **64.26**         | **46.75**                |
>
> **Q1:**
>
> The trace constraint assumed in Theorem 4 is both theoretically meaningful and practically justified. In modern contrastive learning pipelines, feature vectors are typically $\ell_2$-normalized (e.g., in SimCLR and SimSiam), which ensures that the overall energy—captured by the trace of $FF^\top$—is approximately fixed across training batches. This normalization implicitly induces a bounded trace, making our assumption aligned with standard practice. Moreover, since downstream classifiers are scale-invariant, constraining the trace serves to eliminate trivial rescalings of the feature matrix and leads to a well-posed analysis of the spectral structure. We will clarify this motivation more explicitly in the main text.
>
> **Q2:**
>
> We appreciate the reviewer’s suggestion to empirically verify whether the learned feature matrix $F$ approaches orthogonality during training. To this end, we evaluated the _effective rank_ of $F$, which serves as a proxy for spectral flatness and proximity to scaled orthogonality. We conducted this analysis on SpurCIFAR-10 and reported effective rank and the worst-group accuracy (%). Our regularized method achieved higher effective rank than the SimCLR baseline, supporting our theoretical claim that the proposed regularizer promotes more uniform and orthogonal representations. This result will be included in the final version and appendix as further empirical evidence for the connection between spectral uniformity and improved generalization.
>
> | Method               | Effective Rank ↑ | Worst-Group Accuracy (%) ↑ |
> | -------------------- | ---------------- | -------------------------- |
> | SimCLR (baseline)    | 47.31            | 36.11                      |
> | SimCLR + spec (ours) | **58.05**        | **58.82**                  |
>
> This table illustrates how our spectral regularization substantially increases the effective rank of the learned representations, which correlates with improved worst-group performance, aligning with our theoretical motivation.
>
> **Q3:**
>
> Thank you for pointing this out. We agree that the current presentation of Figure 2 may conflate our regularization method with the spectrum truncation ablation, and we will revise the text to improve clarity. To clarify: Figure 2 does not illustrate the effect of our proposed regularizer directly. Rather, it serves as an empirical motivation by showing how post hoc manipulations of the learned feature spectrum (specifically, truncating small singular values and/or flattening the rest) impact downstream performance. As the reviewer correctly notes, our regularizer does not **drop** eigenvalues. Instead, it softly promotes a balanced (i.e., flatter) spectrum via penalizing variance in the eigenvalues. We will make this distinction explicit and revise the wording around "dropping eigenvalues" to avoid confusion. Furthermore, we will refer the reader more clearly to Appendix A, where the separate effects of truncation and flattening are disentangled. We appreciate the feedback and will ensure the final version communicates this more transparently.
>
> Thank you again for your thoughtful and constructive feedback. We believe the additional results and clarifications provide, especially the broader set of baselines (including BYOL, DINO, IFM), and Hard ImageNet [1], and STL-10 [3] evaluation, and the effective rank analysis, address the key concerns regarding the method’s generality, comparisons, and empirical scope. We kindly ask you to consider updating your score accordingly, or let us know if any further questions remain; we would be happy to clarify.
>
> References:
>
> [1] M. Moayeri, S. Singla, and S. Feizi. Hard imagenet: Segmentations for objects with strong spurious cues. NeurIPS, 2022.
>
> [2] K. Hamidieh, H. Zhang, S. Sankaranarayanan, and M. Ghassemi. Views can be deceiving: Improved ssl through feature space augmentation. ArXiv, 2024.
>
> [3] A. Coates, A. Ng, and H. Lee. An analysis of single-layer networks in unsupervised feature learning. AISTATS, 2011.

---

### Official Review · Reviewer_nT26 · 2025-06-30

**Clarity:** 3
**Significance:** 3
**Originality:** 2
**Rating:** 4
**Confidence:** 4

**Summary:**

The paper shows that standard contrastive self-supervised learning concentrates variance in a handful of feature directions; these dominant directions often encode spurious shortcuts, hurting robustness. The authors prove that, under a trace constraint, a uniform eigen-spectrum minimizes expected generalization error across random downstream linear tasks. Guided by this, they add a spectrum-flattening regularizer to SimCLR that penalizes unequal eigenvalues, effectively lifting small singular values and raising the feature matrix’s effective rank. This regularizer improves worst-group and average accuracy on five spurious-correlation benchmarks.

**Questions:**

Please see the weaknesses section above

**Ethical Concerns:**

["NO or VERY MINOR ethics concerns only"]

**Final Justification:**

I thank the authors for their detailed response. Their rebuttal has addressed most of my concerns, so I will increase my score to 4.

**Quality:**

3

**Strengths And Weaknesses:**

**Strengths**
- The paper is very well written, clear and easy to follow.
- The method is grounded in solid theoretical principles --  Theorem 4 rigorously shows that a perfectly uniform eigenspectrum minimises average linear-probe error under a trace constraint.
- Hyper-parameters, optimiser settings, and code snippets for the SVD-based penalty are fully disclosed, making the approach easy to drop into existing contrastive SSL pipelines.

**Weaknesses**

- The proof assumes a square, full-rank feature matrix and purely linear downstream heads. Realistic encoders are tall (feature dim < #samples) and many transfer tasks use non-linear fine-tuning. The paper does not quantify how loose the bound becomes in this common regime, nor whether a flat spectrum remains optimal when features are low-rank.

- Because the penalty uniformly boosts all low-eigenvalue directions, it can inadvertently raise purely noisy components if those directions are not spurious but irrelevant. The paper does not test on datasets where spurious and core signals overlap heavily, so this failure mode is unexamined.

- The authors sub-sample majority groups when training the linear probe (Sec. 6.3). Prior work shows that simple re-balancing alone can recover much of the performance loss caused by group or class imbalance—for instance JTT (Liu et al., 2021), GroupDRO (Sagawa et al., 2020), and “Just Train Twice” (Jiang et al., 2021). Because of this step, the study (i) relies on explicit group labels at probe time, weakening its “label-free” narrative, and (ii) makes it harder to judge how much additional robustness comes from the spectrum-flattening regulariser versus the well-known benefits of balanced sampling. Reporting results with and without sub-sampling—or using natural group imbalance—would clarify the regulariser’s standalone contribution.

- The regularization needs an SVD per mini-batch. At 512dim this is not that significant, but it grows quadratically with dimension. The authors mention possible Hutch++ approximations in Appendix K but does not implement or benchmark them. It is unclear how stable these approximations would be in SSL training.

- All experiments are only conducted on small scale datasets; no evidence is shown for ImageNet-1k, or ImageNet which are now the defacto for SSL evaluations. Further, more exhaustive and large scale benchmarks from WILDS could also be used.

- The paper claims the framework is broadly applicable to representation learning methodologies, but the main experimental results only show it applied to SimCLR. Adding more baselines with the additional regularization will make the experiments section more complete.

- (Minor) Related works section i.e. Section 3 starts abruptly after a nice motivation to the problem in Section 2. It might be better to either place it before Section 2 or towards the end to maintain and coherent flow of information.

---

> ### Author Rebuttal · Authors · 2025-07-31
>
> We thank the reviewer for the constructive and thoughtful evaluation. We appreciate the positive comments on the clarity of the paper, the strength of our theoretical analysis (especially Theorem 4), and the ease of integrating our method into existing pipelines. We're glad the reviewer found our contributions to be both rigorous and practical. Below, we address each of the reviewer's concerns.
>
> **W1:**
>
> As noted, real-world encoders typically produce tall, rank-deficient feature matrices $d \\ll n$ and are often fine-tuned with non-linear heads. To address the reviewer’s concern more directly, we have extended Theorem 4 to the rank-deficient case where $\operatorname{rank}(F) = r < n$, and show that the same conclusion holds: the surrogate loss $\mathcal{L}(F) := \operatorname{tr}((FF^\top)^{\dagger})$ is minimized when the non-zero eigenvalues of $FF^\top$ are uniform. This result strengthens the connection between spectral uniformity and generalization, even in the realistic regime of low-rank encoders. In the proof, we replace the inverse with the Moore–Penrose pseudoinverse $(FF^\top)^{\dagger}$, which is well-defined for all symmetric positive semi-definite matrices and coincides with the true inverse whenever $FF^\top$ is full-rank. Importantly, this replacement preserves the structure of the objective: the trace of the pseudoinverse penalizes low-energy (i.e., low-eigenvalue) directions in the representation space, while ignoring the nullspace where the model has no discriminatory capacity. From a generalization perspective, $Y^\top (FF^\top)^{\dagger} Y$ continues to quantify the expected error for randomly sampled downstream tasks projected onto the span of the learned representation. We have extended our formal theorem.
>
> **Theorem 4** [Rank-deficient Case]
>
> Let $F \in \mathbb{R}^{n \times d}$ be a feature matrix with $\operatorname{rank}(F) = r \le \min(n,d)$, and let $G := FF^\top \in \mathbb{R}^{n \times n}$. Suppose $G$ has eigenvalues $\lambda_1 \ge \lambda_2 \ge \dots \ge \lambda_r > 0$, with $\lambda_{r+1} = \dots = \lambda_n = 0$, and a fixed trace constraint $\sum_{i=1}^r \lambda_i = c$. Then the expected loss
> $
> \mathcal{L}(F) := \mathbb{E}\_{\mathbf{w}, \mathbf{y}} \left[ \| \mathbf{y} - F \mathbf{w} \|^2 \right]
> $
> under the random task model is minimized when the non-zero eigenvalues are uniform:
> $
> \lambda_1 = \dots = \lambda_r.
> $
>
> We also updated **Corollary 3** to reflect the low-rank case, using Moore–Penrose pseudoinverse, in the generalization bound. This ensures consistency with the modified theorem and highlights that our conclusions hold even when $F$ is rank-deficient.
> Due to space limit, we do not restate the full theorem here and the complete theorem and proof will appear in the Appendix upon acceptance.
>
> **Corollary 3** [Rank-deficient Case]
> With probability at least $1 - \delta$, the population loss satisfies
> $
> L\_{\mathcal{D}}(g\_{\mathbf{w}^{(k)}}) := \mathbb{E}\_{(\mathbf{x}, y) \sim \mathcal{D}} \left[ \ell(g_{\mathbf{w}}(\mathbf{x}), y) \right]
> \le \widetilde{O} \left( \sqrt{ \frac{ \mathbf{y}^\top (F F^\top)^{\dagger} \mathbf{y} }{n} } \right),
> $
> where $\mathbf{y} = (y_1, \dots, y_n)^\top$, and $\widetilde{O}$ hides logarithmic factors and dependence on $\delta$.
>
> **W2:**
>
> We thank the reviewer for highlighting a potential failure mode: that our regularizer, by uniformly encouraging spectrum flattening, could inadvertently amplify noisy or irrelevant directions. We emphasize, however, that our method does not enforce a hard constraint on the spectrum. The regularizer is applied softly via a tunable hyperparameter $\alpha$, which balances spectral flattening against the base contrastive objective. In practice, this allows the model to adjust its spectrum to the structure of the data, rather than blindly forcing uniformity.
>
> Regarding the distinction between spurious and core features, we respectfully clarify that in our setting this is not a spatial or subspace-level separation, but an inductive one. Spurious features are defined as those that are _simple and easily learnable_ that are irrelevant to the downstream task at hand. They are picked up early in training due to the optimization bias of neural networks, not because they inhabit distinct eigenvectors. Therefore, the notion of "overlap" between spurious and core signals does not strictly apply. Our goal is not to eliminate spurious features entirely, but to **ensure that underrepresented or task-relevant features are not suppressed** by spectral collapse. By flattening the spectrum moderately, we allow the model to preserve both types of features — especially those that would otherwise be underutilized. We will clarify these conceptual points and the role of the hyperparameter in the revised version.
>
> **W3:**
>
> We thank the reviewer for raising the important question of whether rebalancing strategies could account for the observed robustness gains. We emphasize that our method does not assume access to group labels during self-supervised pretraining, so rebalancing is not performed. However, to test this hypothesis in controlled settings where group information is available, we refer to the systematic evaluation presented in [1]. In Table 2 of their paper, the authors trained SimSiam under three rebalancing strategies applied during pretraining: (i) group-balanced sampling to match the downstream distribution, (ii) downsampling majority groups, and (iii) upsampling minority groups. Across multiple spurious benchmarks including CelebA, MetaShift, CMNIST, SpurCIFAR-10, and Waterbirds, they found that such interventions did _not_ consistently improve worst-group accuracy. These results (in Table 2 of [1]) suggest that simple group reweighting during pretraining does not replicate the robustness benefits observed with spectrum regularization. We will include this discussion in the final version to clarify the distinction.
>
> We thank the reviewer for pointing to Just Train Twice (JTT) [2] and GroupDRO [3] as related work. We were unable to find a separate "Just Train Twice" (JTT) paper by Jiang et al., and would appreciate clarification in case we missed it or if this was intended to refer to Liu et al., 2021. All of these methods operate in a supervised setting and require access to either class labels, group labels, or both during training. GroupDRO explicitly assumes known group annotations and dynamically upweights high-loss groups to optimize worst-group performance. JTT identifies difficult examples by first training a supervised model, then upweighting high-loss samples in a second training pass—relying on access to per-example losses and class labels.
>
> In contrast, our method is applied in the fully self-supervised regime, where neither class nor group labels are available during training. The spectrum-flattening regularizer is designed to improve representation robustness across downstream tasks by encouraging spectral diversity, without relying on any knowledge of group structure or downstream objectives. While JTT and GroupDRO address robustness through explicit supervision or heuristic reweighting, our approach provides a complementary, label-agnostic mechanism that improves generalization even in the absence of downstream supervision. We will clarify this distinction in the related work of the final version.
>
>
> **W4:**
>
> To investigate this further, we conducted an ablation comparing the full SVD against approximate randomized SVD. All evaluations used batch size 256 on SpurCIFAR-10 with our regularized SimCLR.
>
> The results are summarized below:
>
> | SVD Method        | WG Acc. (%) | Avg. Acc. (%) | Effective Rank | GPU Memory (MB) | SVD Time (s) |
> |-------------------|------------------|-------------------|-----------------|------------------|---------------|
> | Full SVD          | 58.34        | 80.34             | 57.93           | 2510         | 0.067        |
> | Approx SVD (k=64) | 54.87            | 80.22             | 57.79           | 2510         | 0.037        |
> | Approx SVD (k=32) | 51.30            | 80.28             | 57.51           | 2483         | 0.035        |
> We observe that:
>
> This suggests that the regularizer is robust to mild approximation errors and that SVD approximation approaches are viable in practice. These experiments demonstrate that the method is practical and scalable to higher dimensions, especially when using rank-limited approximations.
>
> **W5:**
>
> We conducted experiments on Hard ImageNet [4].
> We report the average accuracy on the standard test split (None) for one random seed:
>
> | Method                     | Avg. Accuracy |
> | ------------------------- | -----------------------------: |
> | SimCLR (baseline)         |                          79.06 |
> | SimSiam (baseline)        |                          79.50 |
> | Ours (SimCLR + Reg.)      |                      79.46 |
> | LATETVG [1] |                          78.00 |
>
> **W6**
>
> We thank the reviewer for the suggestion. We have also applied our regularizer to SimSiam, as reported in Appendix H (Tables 5 and 6), demonstrating its compatibility beyond SimCLR.
>
> We believe we have addressed your concerns thoroughly, both theoretically and empirically. We kindly ask you to consider raising your score or let us know if any further clarification is needed.
>
> References:
>
> [1] K. Hamidieh, H. Zhang, S. Sankaranarayanan, and M. Ghassemi. Views can be deceiving: Improved ssl through feature space augmentation. ArXiv, 2024.
>
> [2] E. Z Liu, B. Haghgoo, A. S Chen, A. Raghunathan, P. W. Koh, S. Sagawa, P. Liang, and C. Finn. Just train twice: Improving group robustness without training group information. ICML. PMLR, 2021b.
>
> [3] S. Sagawa, P. W. Koh, T. B Hashimoto, and P. Liang. Distributionally robust neural networks for group shifts: On the importance of regularization for worst-case generalization. ArXiv, 2019.
>
> [4] M. Moayeri, S. Singla, and S. Feizi. Hard imagenet: Segmentations for objects with strong spurious cues. NeurIPS, 2022.

---

> > ### Comment · Reviewer_nT26 · 2025-08-05
> >
> > I thank the authors for their detailed response. Their rebuttal has addressed most of my concerns, so I will increase my score to 4.

---

### Official Review · Reviewer_Dgse · 2025-07-04

**Clarity:** 2
**Significance:** 3
**Originality:** 4
**Rating:** 5
**Confidence:** 5

**Summary:**

This paper shows that the reason why SSRL (self-supervised representation learning) methods learn spurious features, is that the directions which those features are learned dominates the top eigenvalues of the feature covariance $FF^T$. They formalize this by showing that a downstream linear probe (on top of feature representations) learns fastest along large eigenvector directions, so any label information in small eigenvalues is not being used when training the linear probe.  The paper also generalizes this to unknown tasks in Theorem 4, and shows that the expected loss is minimized when all eigenvalues are close to each other. Given this, they propose a Spectrum Flattening Regularizer for SSRL losses and show that it improves downstream worst-group accuracy across a number of tasks.

**Questions:**

Q1: The regularizer flattens the eigenvalues of the mini-batch feature covariance, but Theorem 4 is for the entire dataset. How sensitive is the training signal to the mini-batch estimates of the regularizer? could you plot the eigenvalues that vary across batches during training?

Q2: Do spurious eigenvectors actually vanish when adding the regularizer? Is there a way to quantify this?

Q3: Because eigenvalue estimates improve with larger batches, a fixed regularization $\alpha$ may have different strengths across batches. Does changing the batch size significantly effect tuned $\alpha$?

**Ethical Concerns:**

["NO or VERY MINOR ethics concerns only"]

**Final Justification:**

Keeping my score, since the reviewers have addressed my concerns in the rebuttal.

**Limitations:**

L1: Theorem 4 assumes square full-rank $FF^T$, independent bernoulli labels, and a trace constraint. It is unclear how the “uniform spectrum” optimum generalizes to overparameterized encoders or correlated tasks.

L2: The regulariser flattens the eigenvalues of the batch feature covariance, which can diverge from the full dataset covariance when batches are small or highly correlated.

**Quality:**

3

**Strengths And Weaknesses:**

Strengths:

S1 (theoretical analysis motivated by the empirical observation): The authors start from an empirical observation (that the contrastive features collapse into high variance directions for datasets containing spurious correlations) and which motivates the theory.

S2 (proposed method motivated by the theoretical analysis): The proposed spectrum flattening loss is directly derived from theorem 4's uniform eigenvalue optimum, which makes it principled.

S3: The proposed loss is applicable to any sort of SSRL method that is learning a feature space, and adapting it for other SSRL method would be easy.

Weaknesses:

W1: Paper presentation and writing can be improved. For instance, the problem setup (section 4.1) comes after where these symbols first appear in the paper (section 2.1).

W2: Even though the eigenvalues are flattened using the proposed method, the paper offers no analysis of how the directions itself evolves, which is important for understanding whether spurious vectors are suppressed or rescaled, though the downstream worst-group accuracy improves.

W3: The random-task model and the reduction to $tr[FF^T]^{-1}$ are presented too abruptly, and the paper’s transition from Corollary 3 to Theorem 4 makes it follow in its current form. The “random task” construction that supports Theorem 4 introduces several new vectors and scalars in a single dense paragraph without a guiding figure.

W4: The theorem itself relies on specific assumptions that is unclear if they hold in practice, e.g. independent Bernoulli labels, a uniform $\ell_\infty$, a square full-rank feature matrix, and a trace (rather than Frobenius or spectral-norm) constraint. Especially for overparameterized encoders or correlated downstream tasks.

---

> ### Author Rebuttal · Authors · 2025-07-31
>
> Thank you very much for your detailed and insightful review. We deeply appreciate your recognition of both the theoretical motivation and general applicability of our method, as well as your thoughtful comments on the limitations and open questions.  Below, we address your concerns and questions.
>
> **W2:**
>
> While our proposed regularizer explicitly flattens the _eigenvalues_ of the feature covariance matrix, it does not directly constrain the _eigenvectors_. Nonetheless, the generalization bound presented in Corollary 3 shows that what ultimately matters is the alignment between the label vector $\mathbf{y}$ and the top eigenspaces of $FF^\top$. Thus, the key question is whether some (e.g. spurious) features retain high variance and dominate top eigenspaces, or whether their influence is reduced under the regularizer.
>
> To address this, we plan to visualize the behavior of spurious directions via three plots upon acceptance:
> Specifically, we plan to include the eigenvalue spectrum of $FF^\top$ and the projection magnitudes $\langle \mathbf{y}, \mathbf{v}_i \rangle$ of the label vector onto each eigenvector $\mathbf{v}_i$ for SpurCIFAR-10 trained with SimCLR and trained with SimCLR with our spectral regularizer.
>
> This comparison highlights that in CIFAR-10, label information aligns more strongly with the top eigenspaces, whereas in SpurCIFAR-10, the alignment is weaker due to dominant spurious directions.
>
> **W3:**
>
> Section 4.2 aims to generalize Corollary 3, which applies to a fixed downstream task, to a more realistic _contrastive learning_ setting where the downstream task is _unknown_ at pretraining time. We introduce a random-task framework in which (i) two latent classes $c^+, c^- \in \mathcal{C}$ are sampled, (ii) the classifier vector is $\mathbf{v} = \mathbf{v}\_{c^+} - \mathbf{v}\_{c^-}$, (iii) the labels are generated via the soft classifier, $\mathbb{P}(Y\_i = \pm 1 \mid \mathbf{v}) = \frac{1 \pm (F \mathbf{v})\_i}{2}$, and (iv) the generalization objective becomes $\mathbb{E}\_{\mathbf{v}, Y} \left[ Y^\top (F F^\top)^{-1} Y \right]$.
>
> Theorem 4 shows this is minimized when $FF^\top$ has uniform eigenvalues, directly motivating our spectrum-flattening regularizer. We plan to clarify this section in the camera-ready with a schematic figure of the random-task setup.
>
> **W4:**
>
> We appreciate the reviewer’s thoughtful comments regarding the assumptions underlying Theorem 4. Below, we address each assumption and clarify its motivation or generalization.
>
> **1) Trace constraint.**
> The trace constraint used in Theorem 4 is both theoretically meaningful and practically justified. In modern contrastive learning pipelines—such as SimCLR and SimSiam—it is standard to apply $\ell_2$-normalization to the feature vectors. This ensures that the overall energy of the representation, captured by $\operatorname{tr}(FF^\top)$, remains approximately constant across batches. Thus, the fixed-trace constraint reflects common empirical practice. Furthermore, since downstream classifiers are scale-invariant, normalizing the trace removes trivial rescalings of the feature matrix and makes the spectral analysis of feature representations well-posed. We will make this motivation more explicit in the revised version of the paper.
>
> **2) Uniform $\ell_\infty$ constraint.**
> The $\ell_\infty$ constraint used in our theorem is not essential and can be replaced with any constant $c=c_1, c_2$. The results scale accordingly. In practice, this constraint is naturally satisfied due to common regularization on the feature and classifier norms, such as $\|f_i\|2 \leq c_1$ and $\|v\|_2 \leq c_2$, which are widely adopted in deep contrastive learning setups. We used the uniform $\ell_\infty$ constraint for analytical clarity, and the bound remains valid under more general norm bounds.
>
> **3) Independent Bernoulli labels.**
> Our analysis is designed to **measure the quality of features learned by contrastive learning**. The key intuition is that *good representations* should support *easy downstream classification*, typically through a linear model. To formalize this, we consider a stylized label generation process: given feature $f_i$ and two contrast vectors $v^+$ and $v^-$, the label $+1$ is assigned with probability proportional to $f_i^\top v^+ - f_i^\top v^-$, and vice versa. This assumption allows for clean analysis and directly evaluates whether the learned features are linearly discriminative. It serves as a principled surrogate for linear separability, a widely accepted proxy for representation quality in the contrastive learning literature (e.g., SimCLR, MoCo).
>
> **4) Full-rank assumption.**
> While our theorem assumes a square, full-rank feature matrix $F$, this is a simplifying assumption and not fundamental to the results. The main requirement is that the Gram matrix $FF^\top$ is full-rank, a condition commonly used in analyses of overparameterized models (e.g., [Arora et al., 2019]) to ensure that the optimization landscape allows perfect fitting. This condition is satisfied in typical **overparameterized** regimes where $F \in \mathbb{R}^{n \times d} \) with \( d \geq n$, such as in contrastive encoders with high-dimensional features.
>
> In the underparameterized case $d \ll n$, our analysis can be naturally extended via the eigendecomposition $FF^\top = \sum \lambda_i v_i v_i^\top$, assuming that the label vector $y$ lies in the span of the eigenvectors with non-zero eigenvalues (i.e., $y^\top v_i = 0$ whenever $\lambda_i = 0$). **Under this milder assumption**, the bound in Corollary 3 becomes:
> $
> y^\top \left( \sum_{i: \lambda_i \ne 0} \frac{1}{\lambda_i} v_i v_i^\top \right) y,
> $
> which reflects the contribution of the label only along the directions where the feature matrix has support.
>
> **For Theorem 4, this implies that the spectrum must be uniform (constant) over the first $r$ non-zero eigenvalues, and zero elsewhere**, in order to achieve optimality. This provides a clear geometric interpretation: the features are well-structured if they uniformly capture the relevant directions in the label space, while ignoring degenerate components. This insight remains informative even when the feature matrix is low-rank, and highlights how our results apply beyond the strictly full-rank case.
>
> We appreciate the reviewer’s concerns and hope this response clarifies the role and flexibility of each assumption. Our goal is to provide a theoretical lens on feature quality in contrastive learning, and while our assumptions are stylized for analytical clarity, they are either standard in related theoretical work or adaptable to broader regimes. We will explicitly incorporate these clarifications and generalizations in the final version.
>
> **Q1:**
>
> We thank the reviewer for raising this important point. While Theorem 4 analyzes the full-dataset spectrum of the feature matrix $F$, our regularizer operates on mini-batch estimates of $FF^\top$. To assess the sensitivity of the training signal to this approximation, we will empirically analyze the spectrum across mini-batches during training.
>
> This will be similar in spirit to the visualization we proposed in **W2**, where we plan to plot the eigenvalue spectrum of $FF^\top$ and the projection magnitudes $\langle \mathbf{y}, \mathbf{v}_i \rangle$ of the label vector onto each eigenvector $\mathbf{v}_i$, computed for multiple mini-batches during training. These plots will help quantify whether the flattening effect is consistent across batches and whether eigendirections are suppressed or simply rescaled.
>
> We will include these visualizations upon acceptance.
>
> **Q2:**
>
> The spurious eigenvectors do not vanish when applying the proposed regularizer. Instead, their relative spectral dominance is attenuated, allowing other directions—potentially aligned with core or task-relevant features—to become more prominent. The goal of our approach is not to suppress or discard any specific directions, but rather to mitigate over-reliance on a small number of high-variance components. This aligns with our Theorem 4, which suggests that a more uniform spectrum promotes better generalization across downstream tasks when their identity is unknown at pretraining time. We emphasize that spurious directions may still contain useful information for certain downstream tasks. Since the task is not known during representation learning, it is desirable to maintain a feature space that preserves all directions while distributing variance more evenly. The regularizer achieves this by encouraging the spectrum of $FF^\top$ to become flatter, thereby reducing the disproportionate influence of dominant eigenvectors without erasing them. To empirically assess this effect, we will (upon acceptance) include visualizations of the eigenvalue spectrum and the corresponding label projection magnitudes $\langle \mathbf{y}, \mathbf{v}_i \rangle$.
>
> **Q3:**
>
> To investigate this, we conducted an experiment on SpurCIFAR-10 using SimCLR trained with our regularizer, fixing the regularization weight to $\alpha = 0.01$ and varying the batch size. Each setting was run with a single random seed to isolate batch-size effects.
>
> | **Batch Size** | **Avg. Acc. (%)** | **WG. Acc. (%)** | **Eff. Rank** |
> | -------------- | -------------------- | ---------------------------- | ------------------ |
> | 512            | 77.93                 | 55.77                         | 30.1               |
> | 256            | 80.1                 | 58.82                         | 58.05               |
> | 128            | 80.94                 | 60.31                         | 68.02               |
> | 64             | 81.74                 | 54.16                         | 48.86               |
>
> We will include an additional ablation study on the effect of varying batch sizes upon acceptance.
>
> We hope that the revisions and additional discussions have addressed your concerns. If any issues remain unclear or open, we’d be happy to provide further clarification.

---

> > ### Comment · Reviewer_Dgse · 2025-08-05
> >
> > Thank you for your response. My initial questions regarding the paper are mostly addressed in the rebuttal.
> > You may want to include your response to W3 and W4 in a future version of the paper. I am keeping my score.

---

### Official Review · Reviewer_EvGa · 2025-07-04

**Clarity:** 3
**Significance:** 3
**Originality:** 3
**Rating:** 5
**Confidence:** 4

**Summary:**

This paper addresses the challenge of model reliance on spurious correlations in contrastive self-supervised learning. The authors propose a spectral regularization method that encourages the eigenvalues of the feature covariance matrix to be close to the largest eigenvalue, promoting a higher-rank representation. The paper provides a theoretical perspective linking downstream generalization error to the rank of the feature covariance matrix. Empirically, the method is applied to SimCLR and shows improved performance on benchmarks with spurious correlations, outperforming alternative approaches such as LateTVG on metrics such as worst-group and average accuracy.

**Questions:**

1. **Related Work on Rank-Promoting Regularization.**
The related works section would benefit from an additional discussion of related literature that explores regularization techniques aimed at improving the rank or diversity of learned representations such as Barlow Twins, VICReg, Whitening-based approaches. Including these connections would help contextualize the proposed method and clarify its novelty relative to existing spectrum- or correlation-based regularizers.

2. **Computational Overhead of Spectral Regularization.** The paper would benefit from quantifying the training and memory overhead introduced by the spectral regularizer, especially relative to other methods such as VICReg and Barlow Twins. Including this analysis in the main manuscript (rather than supplementary material) would improve transparency and help practitioners assess the cost-benefit tradeoff.

3. **Sensitivity Analysis of the Regularization Strength Hyperparameter.**
The method introduces a hyperparameter that controls the strength of the spectral regularization. It would be helpful to include a sensitivity analysis showing how performance varies with different settings of this hyperparameter. This would provide insight into the robustness of the method and guide practitioners on how to tune it effectively.

**Ethical Concerns:**

["NO or VERY MINOR ethics concerns only"]

**Final Justification:**

I have increased my score during the rebuttal since the authors have adequately addressed my concerns .

**Limitations:**

Yes, in the Appendix.

**Paper Formatting Concerns:**

None.

**Quality:**

3

**Strengths And Weaknesses:**

Strengths:
- This paper proposes a regularization method that encourages a high-rank feature covariance matrix that is supported by both theoretical analysis and empirical results, and leads to improved downstream performance.
- The proposed regularization is architecture-agnostic.
- The experiments report confidence intervals over five seeds, enhancing the statistical reliability of the findings.
- The authors provide pseudocode and hyperparameter values which can enhance reproducibility.
- In the questionnaire, they also mention that implementation of the method would be provided.

Room for improvement:
- I believe the paper could benefit from including information about related work on regularization, the computational cost of the proposed method, and sensitivity analysis, some of which are available in the appendix. Please see details in the following section.

---

> ### Author Rebuttal · Authors · 2025-07-30
>
> Thank you very much for your detailed and thoughtful review. We appreciate your recognition of the theoretical and empirical strengths of our work, as well as the clarity, reproducibility, and architectural generality of the proposed regularizer. We address the specific concerns below.
>
> 1. **Related Work on Rank-Promoting Regularization.**
>
> We thank the reviewer for highlighting the importance of better situating our work among existing rank-promoting approaches in self-supervised learning. We have updated the related work section to explicitly include comparisons to methods such as Barlow Twins, VICReg, whitening-based regularization, and IFM.
>
> **Rank-Promoting Regularization.** Several recent methods explicitly incorporate regularizers to encourage more diverse, high-rank representations in self-supervised learning. For example, Barlow Twins [1] introduces a redundancy-reduction loss that drives the cross-correlation matrix of twin network embeddings toward the identity, thereby decorrelating feature dimensions and minimizing redundancy. Similarly, VICReg [2] adds a variance preservation term and a covariance penalty to maintain per-dimension variance while pushing off-diagonal covariances toward zero, thus preventing informational collapse by decorrelating features. Whitening-based approaches go even further by forcing the entire feature covariance to match an identity matrix (full whitening), which is equivalent to enforcing a full-rank embedding space and comes with theoretical guarantees against dimensional collapse [3]. Beyond these objectives on the representation statistics, other techniques like Implicit Feature Modification (IFM)  [4] actively perturb training examples in feature space to ensure the encoder utilizes a broader set of features, reducing reliance on any single dominant “shortcut” and promoting greater feature diversity. All of these strategies underline the importance of promoting feature diversity and high effective dimensionality in learned representations. Our proposed approach aligns with this principle but differs by emphasizing a balanced eigenspectrum (neither overly flat nor too concentrated), aiming to mitigate spurious feature dominance without the extremes of full whitening or low-rank collapse, thereby supporting both robust and transferable features.
>
> 2. **Computational Overhead of Spectral Regularization.**
>
> Many self-supervised learning methods, such as Barlow Twins [1] and VICReg [2], reduce feature redundancy by computing and differentiating through $d\times d$ matrices. Whitening-based methods, like ZCA whitening [3], are even more computationally expensive due to requiring an eigendecomposition complexity.
> In contrast, our spectral flattening regularizer only relies on computing the singular values of the feature matrix $F$.
>
> | **Method**                          | **Forward + Backward Cost** |
> | ----------------------------------- | --------------------------- |
> | Barlow Twins (Zbontar et al., 2021) | $\mathcal{O}(nd^2 + d^2)$   |
> | VICReg (Bardes et al., 2022)        | $\mathcal{O}(nd^2 + d^2)$   |
> | Whitening (Ermolov et al., 2021)    | $\mathcal{O}(nd^2 + d^3)$   |
> | **Spectral Flattening (Ours)**      | $\mathcal{O}(nd^2)$         |
>
> We provide a comprehensive explanation in the Appendix J and would be glad to include it in the main paper for improved clarity.
>
> 3. **Sensitivity Analysis of the Regularization Strength Hyperparameter.**
>
> We thank the reviewer for requesting a sensitivity analysis, which we now provide below. We evaluated the performance of our method on the MetaShift benchmark using different values of the spectral regularization strength $\alpha$, using a fixed seed. The results are presented in the following table:
>
> | Regularization Strength \$\alpha\$ | Average Accuracy (%) |
> | :--------------------------------: | :------------------: |
> |                0.001               |         76.40        |
> |               0.0025               |         77.53        |
> |                0.005               |         80.34.       |
> |                0.010               |         76.97        |
> |                0.025               |         75.28        |
> |                0.050               |         77.53        |
> |                0.100               |         76.40        |
>
> We observe that while the method shows some sensitivity to the hyperparameter, it maintains strong performance in a broad range, particularly from $\alpha = 0.005$ to $\alpha = 0.05$. The best performance was achieved at $\alpha = 0.005$. This provides a practical default for tuning.
>
> We are currently running additional experiments across multiple seeds to confirm robustness, and we will include those upon acceptance.
>
> Your comments on related work, computational overhead, and sensitivity analysis are especially valuable and have helped us improve the presentation and scope of the paper. If any concerns remain or further clarification is needed, we would be happy to address them. Otherwise, we kindly ask you to consider increasing your score.
>
> References:
>
> [1] Jure Zbontar, Li Jing, Ishan Misra, Yann LeCun, and Stéphane Deny. Barlow twins: Self-supervised learning via redundancy reduction. In International conference on machine learning, pages 12310–12320. PMLR, 2021.
>
> [2] Adrien Bardes, Jean Ponce, and Yann LeCun. Vicreg: Variance-invariance-covariance regularization for self-supervised learning. arXiv preprint arXiv:2105.04906, 2021.
>
> [3] Aleksandr Ermolov, Aliaksandr Siarohin, Enver Sangineto, and Nicu Sebe. Whitening for self-supervised representation learning. In International conference on machine learning, pages 3015–3024. PMLR, 2021.
>
> [4] Joshua Robinson, Li Sun, Ke Yu, Kayhan Batmanghelich, Stefanie Jegelka, and Suvrit Sra. Can contrastive learning avoid shortcut solutions? Advances in neural information processing systems, 34:4974–4986, 2021.

---

> > ### Comment · Reviewer_EvGa · 2025-08-05
> > **Thank you and increasing score**
> >
> > I thank the authors for their thoughtful and thorough responses. They have adequately addressed my concerns regarding related work, computational overhead, and sensitivity analysis. The additional results and clarifications strengthen the paper’s contribution and improve its clarity. I am updating my score to reflect these improvements.

---

### Note · Authors · 2025-08-12

Dear Reviewers,

We thank you for your constructive feedback and for recognizing the theoretical grounding, clarity, and empirical contributions of our work. In our rebuttal, we have carefully addressed the raised concerns, further strengthening both the theoretical and practical relevance of our approach.

Best regards,

The Authors

---

### Decision · Program_Chairs · 2025-09-17

**Decision:**

Accept (poster)

**Comment:**

This paper proposes a method that uses spectral regularization to avoid the spurious features that are often acquired by contrastive learning. After empirically demonstrating a relationship between the learning of spurious features during contrastive learning and the tendency of the feature matrix to become low-rank, the authors theoretically show that training a linear predictor on a fixed feature space is strongly influenced by such low-rank structure. Building on this theory, they propose a spectral regularization scheme and empirically demonstrate that it improves performance on downstream tasks.

The paper offers solid theoretical claims regarding the impact of the spectrum of the feature matrix, and, grounded in that theory, puts forward an architecture-agnostic method. Hyperparameters and other details are reported with sufficient clarity to enable reproducibility. Concerns from reviewers regarding related work and the numerical evaluation were addressed through the rebuttal. For these reasons, I recommend acceptance.